# SOPE: Spectrum of Off-Policy Estimators

**Christina J. Yuan**
University of Texas at Austin
cjyuan@cs.utexas.edu

**Yash Chandak**
University of Massachusetts
ychandak@cs.umass.edu

**Stephen Giguere**
University of Texas at Austin
sgiguere@cs.utexas.edu

**Philip S. Thomas**
University of Massachusetts
pthomas@cs.umass.edu

**Scott Niekum**
University of Texas at Austin
sniekum@cs.utexas.edu

## Abstract

Many sequential decision making problems are high-stakes and require off-policy evaluation (OPE) of a new policy using historical data collected using some other policy. One of the most common OPE techniques that provides unbiased estimates is trajectory based importance sampling (IS). However, due to the high variance of trajectory IS estimates, importance sampling methods based on state-action visitation distributions (SIS) have recently been adopted. Unfortunately, while SIS often provides lower variance estimates for long horizons, estimating the state-action distribution ratios can be challenging and lead to biased estimates. In this paper, we present a new perspective on this bias-variance trade-off and show the existence of a spectrum of estimators whose endpoints are SIS and IS. Additionally, we also establish a spectrum for doubly-robust and weighted version of these estimators. We provide empirical evidence that estimators in this spectrum can be used to trade-off between the bias and variance of IS and SIS and can achieve lower mean-squared error than both IS and SIS.

## 1 Introduction

Many sequential decision making problems, such as automated health-care, robotics, and online recommendations are high-stakes in terms of health, safety, or finance [Liao et al., 2020, Brown et al., 2020, Theocharous et al., 2020]. For such problems, collecting new data to evaluate the performance of a new decision rule, called an evaluation policy $\pi_e$, may be expensive or even dangerous if $\pi_e$ results in undesired outcomes. Therefore, one of the most important challenges in such problems is the estimation of the performance $J(\pi_e)$ of the policy $\pi_e$ *before its deployment*.

Many off-policy evaluation (OPE) methods enable estimation of $J(\pi_e)$ with historical data collected using an existing decision rule, called a behavior policy $\pi_b$. One popular OPE technique is trajectory-based importance sampling (IS) [Precup, 2000]. While this method is both non-parametric and provides unbiased estimates of $J(\pi_e)$, it suffers from the *curse of horizon* and can have variance exponential in the horizon length [Jiang and Li, 2016, Guo et al., 2017]. To mitigate this problem, recent methods use stationary distribution importance sampling (SIS) to adjust the *stationary distribution* of the Markov chain induced by the policies, instead of the individual trajectories [Liu et al., 2018, Gelada and Bellemare, 2019, Nachum and Dai, 2020]. This requires (parametric) estimation of the ratio between the stationary distribution induced by $\pi_e$ and $\pi_b$. Unfortunately, estimating this ratio accurately can require *unverifiably* strong assumptions on the parameters [Jiang and Huang, 2020], and often requires solving non-trivial min-max saddle point optimization problems [Yang et al., 2020]. Consequently, if the parameterization is not rich enough, then it may not be possible to represent the distribution ratios accurately, and when using rich function approximators (such as neural networks) then the optimization procedure may get stuck in sub-optimal saddle points.

35th Conference on Neural Information Processing Systems (NeurIPS 2021).

In practice, these challenges can introduce error when estimating the distribution ratio, potentially leading to arbitrarily biased estimates of $J(\pi_e)$, even when an infinite amount of data is available.

In this work, we present a new perspective on the bias-variance trade-off for OPE that bridges the unbiasedness of IS and the often lower variance of SIS. Particularly, we show that

- There exists a *spectrum* of OPE estimators whose end-points are IS and SIS, respectively.
- Estimators in this spectrum can have lower mean-squared error than both IS and SIS.
- This spectrum can also be established for doubly-robust and weighted version of IS and SIS.

In Sections 3 and 4 we show how trajectory-based and distribution-based methods can be combined. The core idea establishing the existence of this spectrum relies upon first splitting individual trajectories into two parts and then computing the probability of the first part using SIS and IS for the latter. In Section 5, we introduce weighted and doubly-robust extensions of the spectrum. Finally, in Section 6, we present empirical case studies to highlight the effectiveness of these new estimators.

## 2 Background

**Notation:** A Markov decision process (MDP) is a tuple $(\mathcal{S}, \mathcal{A}, r, T, \gamma, d_1)$, where $\mathcal{S}$ is the state set, $\mathcal{A}$ is the action set, $r$ is the reward function, $T$ is the transition function, $\gamma$ is the discounting factor, and $d_1$ is the initial state distribution. Although our results extend to the continuous setting, for simplicity of notation we assume that $\mathcal{S}$ and $\mathcal{A}$ are finite. A policy $\pi$ is a distribution over $\mathcal{A}$, conditioned on the state. Starting from initial state $S_1 \sim d_1$, policy $\pi$ interacts with the environment iteratively by sampling action $A_t$ at every time step $t$ from $\pi(\cdot|S_t)$. The environment then produces reward $R_t$ with the expected value $r(S_t, A_t)$, and transitions to the next state $S_{t+1}$ according to $T(\cdot|S_t, A_t)$. Let $\boldsymbol{\tau} := (S_1, A_1, R_1, S_2, ..., S_L, A_L, R_L)$ be the sequence of random variables corresponding to a trajectory sampled from $\pi$, where $L$ is the horizon length. Let $p_\pi$ denote the distribution of $\boldsymbol{\tau}$ under $\pi$.

**Problem Statement:** The performance of any policy $\pi$ is given by its value defined by the expected discounted sum of rewards $J(\pi) := \mathbf{E}_{\boldsymbol{\tau} \sim p_\pi}[\sum_{t=1}^{L} \gamma^{t-1} R_t]$. The infinite horizon setting can be obtained by letting $L \to \infty$. In general, for any random variable, we use the superscript of $i$ to denote the trajectory associated with it. The goal of the off-policy policy evaluation (OPE) problem is to estimate the performance $J(\pi_e)$ of an evaluation policy $\pi_e$ using only a batch of historical trajectories $D := \{\tau^i\}_{i=1}^{m}$ collected from a different behavior policy $\pi_b$. This problem is challenging because $J(\pi_e)$ must be estimated using only observational, off-policy data from the deployment of a different behavior policy $\pi_b$. Additionally, this problem might not be feasible if the data collected using $\pi_b$ is not informative about the outcomes possible under $\pi_e$. Therefore, to make the problem tractable, we make the following standard support assumption, which implies that any outcome possible under $\pi_e$ also has non-zero probability of occurring under $\pi_b$.

**Assumption 1.** *For all $s \in \mathcal{S}$ and $a \in \mathcal{A}$, the ratio $\frac{\pi_e(a|s)}{\pi_b(a|s)} < \infty$.*

**Trajectory-Based Importance Sampling:** One of the earliest methods for estimating $J(\pi_e)$ is trajectory-based importance sampling. This method corrects the difference in distribution of $\pi_b$ and $\pi_e$ by re-weighting the trajectories from $\pi_b$ in $D$ by the probability ratio of the trajectory under $\pi_e$ and $\pi_b$, i.e. $\frac{p_{\pi_e}(\tau)}{p_{\pi_b}(\tau)} = \prod_{t=1}^{L} \frac{\pi_e(A_t|S_t)}{\pi_b(A_t|S_t)}$. Let the single-step action likelihood ratio be denoted $\boldsymbol{\rho}_t := \frac{\pi_e(A_t|S_t)}{\pi_b(A_t|S_t)}$ and the likelihood ratio from steps $j$ to $k$ denoted $\boldsymbol{\rho}_{j:k} := \prod_{t=j}^{k} \boldsymbol{\rho}_t$. The full-trajectory importance sampling (IS) estimator and the per-decision importance sampling (PDIS) estimator [Precup, 2000] can then be defined as:

$$\text{IS}(D) := \frac{1}{m} \sum_{i=1}^{m} \rho_{1:L}^i \sum_{t=1}^{L} \gamma^{t-1} R_t^i, \qquad \text{PDIS}(D) := \frac{1}{m} \sum_{i=1}^{m} \sum_{t=1}^{L} \gamma^{t-1} \rho_{1:t}^i R_t^i,$$

It was shown by Precup [2000] that under Assumption 1, $\text{IS}(D)$ and $\text{PDIS}(D)$ are unbiased estimators of $J(\pi_e)$. That is, $J(\pi_e) = \mathbf{E}_{\boldsymbol{\tau} \sim p_{\pi_b}}[\text{IS}(\boldsymbol{\tau})] = \mathbf{E}_{\boldsymbol{\tau} \sim \pi_b}[\text{PDIS}(\boldsymbol{\tau})]$. Unfortunately, however, both IS and PDIS directly depend on the product of importance ratios and thus can often suffer from exponentially high-variance in the horizon length $L$, known as the "curse of horizon" [Jiang and Li, 2016, Guo et al., 2017, Liu et al., 2018].

**Distribution-Based Importance Sampling:** To eliminate the dependency on trajectory length, recent works apply importance sampling over the state-action space rather than the trajectory space. For any policy $\pi$, let $d_t^\pi$ denote the induced state-action distribution at time step $t$, i.e. $d_t^\pi(s,a) = p_\pi(S_t = s, A_t = a)$. Let the average state-action distribution be $d^\pi(s,a) := (\sum_{t=1}^L \gamma^{t-1} d_t^\pi(s,a))/(\sum_{t=1}^L \gamma^{t-1})$. This gives the likelihood of encountering $(s,a)$ when following policy $\pi$ and averaging over time with $\gamma$-discounting. Let $(S,A) \sim d^\pi$ and $(S,A) \sim d_t^\pi$ denote that $(S,A)$ are sampled from $d^\pi$ and $d_t^\pi$ respectively. The performance of $\pi_e$ can be expressed as,

$$J(\pi_e) = \mathbf{E}_{\boldsymbol{\tau} \sim p_{\pi_e}}\left[\sum_{t=1}^L \gamma^{t-1} R_t\right] = \sum_{s,a}\sum_{t=1}^L \gamma^{t-1} d_t^{\pi_e}(s,a)r(s,a) = \left(\sum_{t=1}^L \gamma^{t-1}\right)\sum_{s,a} d^{\pi_e}(s,a)r(s,a)$$

$$\overset{(a)}{=} \left(\sum_{t=1}^L \gamma^{t-1}\right)\sum_{s,a} d^{\pi_b}(s,a)\frac{d^{\pi_e}(s,a)}{d^{\pi_b}(s,a)}r(s,a) = \sum_{s,a}\sum_{t=1}^L \gamma^{t-1} d_t^{\pi_b}(s,a)\frac{d^{\pi_e}(s,a)}{d^{\pi_b}(s,a)}r(s,a),$$

$$= \mathbf{E}_{\boldsymbol{\tau} \sim p_{\pi_b}}\left[\sum_{t=1}^L \gamma^{t-1}\frac{d^{\pi_e}(S_t,A_t)}{d^{\pi_b}(S_t,A_t)}R_t\right],$$

where (a) is possible due to Assumption 1. Using this observation, recent works have considered the following stationary-distribution importance sampling estimator [Liu et al., 2018, Yang et al., 2020, Jiang and Huang, 2020],

$$\text{SIS}(D) := \frac{1}{m}\sum_{i=1}^m\sum_{t=1}^L \gamma^{t-1} w(S_t^i, A_t^i)R_t^i,$$

where $w(s,a) := \frac{d^{\pi_e}(s,a)}{d^{\pi_b}(s,a)}$ is the distribution correction ratio. Notice that $\text{SIS}(\tau)$ marginalizes over the product of importance ratios $\rho_{1:t}$, and thus can help in mitigating variance's dependence on horizon length for PDIS and IS estimators. When an unbiased estimate of $w$ is available, then $\text{SIS}(\tau)$ is also an unbiased estimator, i.e., $\mathbf{E}_{\boldsymbol{\tau} \sim \pi_b}[\text{SIS}(\tau)] = J(\pi_e)$. Unfortunately, such an estimate of $w$ is often not available. For large-scale problems, parametric estimation $w$ is required in practice and we replace the true density ratios $w$ with an estimate $\hat{w}$. However, estimating $w$ accurately may require both a non-verifiable strong assumption on the parametric function class, and global solution to a non-trivial min-max optimization problem [Jiang and Huang, 2020, Yang et al., 2020]. When these conditions are not met, SIS estimates can be arbitrarily biased, even when an infinite amount of data is available.

# 3 Combining Trajectory-Based and Density-Based Importance Sampling

Trajectory-based and distribution-based importance sampling methods are typically presented as alternative methods of applying importance sampling for off-policy evaluation. However, in this section we show that the choice of estimator is not binary, and these two styles of computing importance weights can actually be combined into a single importance sampling estimate. Furthermore, using this combination, in the next section, we will derive a spectrum of estimators that allows interpolation between the trajectory-based PDIS and distribution-based SIS, which will often allow us trade-off between the strengths and weaknesses of these methods.

Intuitively, trajectory-based and distribution-based importance sampling provide two different ways of correcting the distribution mismatch under the evaluation and behavior policies. Trajectory-based importance sampling corrects the distribution mismatch by examining how likely policies are to take the same sequence of actions and thus applies the action likelihood ratio as the correction term. Distribution-based importance sampling corrects the mismatch by how likely policies are to visit the same state and action pairs—while remaining agnostic to *how* they arrived—and applies the distribution ratio as the importance weight. However, using distribution ratio and action likelihood ratio correction terms are not mutually exclusive, and one can draw on both types of correction terms to derive combined estimators.

To build intuition for why likelihood ratios and distribution ratios can naturally be combined, we consider the two rooms domain shown in Figure 3. In this example, there are two policies $\pi_b, \pi_e$

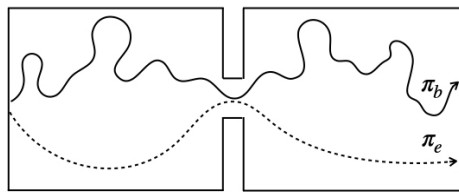

Figure 1: Illustration of two room domain. The domain consists of two rooms, the left room and the right room separated by a connecting door. $\pi_b$ and $\pi_e$ are two different policies that move from the left room to the right room. Note that, although $\pi_b$ and $\pi_e$ have two different behaviors in the left room and right room, both pass through the connecting door.

which have different strategies for navigating from the first room to the second room. Note that while the behavior of the two policies are very different in the left room, both policies must pass through the connecting door to get to the right room at some point in time. Conditioning on having passed through the connecting door at a point in time, all parts of the trajectory that occur in the right room are independent from what has occurred in the left room by the Markov property. Thus, when considering a reward $R_t$ that occurs in the right room, it is natural to consider the probability of reaching the door and then the probability of the action sequence policy in the right room under each policy.

Now, we formalize this intuition and show how trajectory-based and density-based importance sampling can be combined in the same estimator. Given a trajectory $\tau$, we can consider $(S_z, A_z)$, the state and action at time $z$ in the trajectory. By conditioning on $(S_z, A_z)$, trajectory $\tau$ can be separated into two conditionally independent partial trajectories $\tau_{0:z}$ and $\tau_{z+1,L}$ by the Markov property. Since the segments of $\tau$ before and after time $z$ are conditionally independent, then $\rho_{1:z}$, the likelihood ratio for the trajectory before time $z$, is conditionally independent from $\rho_{z+1:L}$ and from $R_t$ for all $t \geq z$. Formally, let $(S_z, A_z) \sim d_z^{\pi_b}$, then,

$$J(\pi_e) = \mathbf{E}_{\tau \sim p_{\pi_b}}[\text{PDIS}(\tau)] = \mathbf{E}_{\tau \sim p_{\pi_b}}\left[\sum_{t=1}^{L} \gamma^{t-1} \rho_{1:t} R_t\right]$$

$$= \mathbf{E}_{\tau \sim p_{\pi_b}}\left[\sum_{t=1}^{z} \gamma^{t-1} \rho_{1:t} R_t\right] + \mathbf{E}_{\substack{(S_z, A_z) \\ \sim d_z^{\pi_b}}}\left[\mathbf{E}_{\tau \sim p_{\pi_b}}\left[\sum_{t=z+1}^{L} \gamma^{t-1} \rho_{1:z} \rho_{z+1:t} R_t \middle| S_z, A_z\right]\right]$$

$$= \mathbf{E}_{\tau \sim p_{\pi_b}}\left[\sum_{t=1}^{z} \gamma^{t-1} \rho_{1:t} R_t\right] + \mathbf{E}_{\substack{(S_z, A_z) \\ \sim d_z^{\pi_b}}}\left[\sum_{t=z+1}^{L} \gamma^{t-1} \mathbf{E}_{\tau \sim p_{\pi_b}}[\rho_{1:z}|S_z, A_z] \, \mathbf{E}_{\tau \sim \pi_b}[\rho_{z+1:t} R_t | S_z, A_z]\right]$$

$$\stackrel{(a)}{=} \mathbf{E}_{\tau \sim p_{\pi_b}}\left[\sum_{t=1}^{z} \gamma^{t-1} \rho_{1:t} R_t\right] + \mathbf{E}_{\substack{(S_z, A_z) \\ \sim d_z^{\pi_b}}}\left[\sum_{t=z+1}^{L} \gamma^{t-1} \frac{d_z^{\pi_e}(S_z, A_z)}{d_z^{\pi_b}(S_z, A_z)} \mathbf{E}_{\tau \sim p_{\pi_b}}\left[\rho_{z+1:t} R_t \middle| S_z, A_z\right]\right]$$

$$= \mathbf{E}_{\tau \sim p_{\pi_b}}\left[\sum_{t=1}^{z} \gamma^{t-1} \rho_{1:t} R_t + \sum_{t=z+1}^{L} \gamma^{t-1} \frac{d_z^{\pi_e}(S_z, A_z)}{d_z^{\pi_b}(S_z, A_z)} \rho_{z+1:t} R_t\right]. \tag{1}$$

where (a) follows from the following Property 1, which states that the expected value of product likelihood ratios $\rho_{1:z}$ conditioned on $(S_z, A_z)$ is equal to the time-dependent state-action distribution ratio for $(S_z, A_z)$. We provide a detailed proof of Property 1 in Appendix A.

**Property 1** ([Liu et al., 2018]). *Under Assumption 1,* $\mathbf{E}_{\tau \sim p_{\pi_b}}[\rho_{1:t}|S_t = s, A_t = a] = \frac{d_t^{\pi_e}(s,a)}{d_t^{\pi_b}(s,a)}$.

Observe that Eq (1) is indexed by time $z$. Intuitively, $z$ can be thought of as the time to switch from using distribution ratios to action likelihood ratios in the importance weight. Specifically, the distribution ratios are used to estimate the probability of being in state $S_z$ and taking action $A_z$ at time $z$ and action likelihood ratios are used to correct for the probability of actions taken after time $z$. Further observe that $z$ does not have to be a fixed constant—$z(t)$ can be a function of $t$ so that each reward in the trajectory $R_t$ can utilize a different switching time. In the next section, we show that by using a function $z(t)$ that allows the switching time to be time-dependent, we are able to further marginalize over time and create an estimator that interpolates between *average* state-action distribution ratios $w(s, a) = \frac{d^{\pi_e}(s,a)}{d^{\pi_b}(s,a)}$, rather than time-dependent distribution ratios $\frac{d_t^{\pi_e}(s,a)}{d_t^{\pi_b}(s,a)}$.

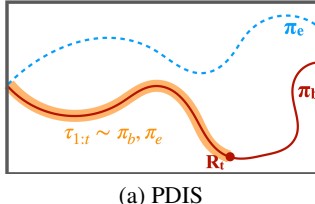
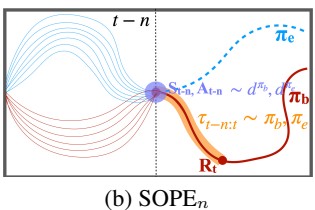
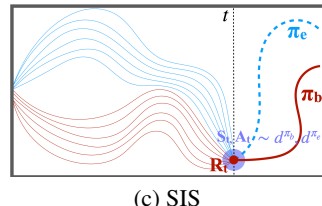

| (a) PDIS | (b) SOPE$_n$ | (c) SIS |

Figure 2: Illustrations of the PDIS, SOPE$_n$ and SIS estimators. The dotted blue line represents an example trajectory drawn from $\pi_e$, and the solid red line represents an example trajectory from $\pi_b$. All three importance sampling methods work by re-weighting each reward $R_t$ in the trajectory from $\pi_b$. (a) Trajectory-based PDIS works by re-weighting each reward by $\frac{p_{\pi_e}(\tau_{1:t})}{p_{\pi_b}(\tau_{1:t})}$, the probability ratio of the sub-trajectory leading up to $R_t$ under the $\pi_b$ and $\pi_e$, respectively. This factors into $\rho_{1:t}$, the product of $t$ action likelihood ratios. (c) Distribution-based SIS considers the probability of encountering $(S_t, A_t)$ under $\pi_e$ and $\pi_b$, and re-weights $R_t$ by $\frac{d^{\pi_e}(S_t, A_t)}{d^{\pi_b}(S_t, A_t)}$, (b) SOPE$_n$ combines trajectory and distribution importance sampling weights by considering the probability of each policy visiting $(S_{t-n}, A_{t-n})$, the state-action pair $n$ steps in the past, and additionally the probability of the sub-trajectory $\tau_{t-n+1:t}$ from $n$ steps in the past to $t$. Thus, SOPE$_n$ re-weights $R_t$ by $\frac{d^{\pi_e}(S_{t-n}, A_{t-n})}{d^{\pi_b}(S_{t-n}, A_{t-n})}\rho_{t-n+1:t}$.

## 4 Bias-Variance Trade-off using $n$-step Interpolation Between PDIS and SIS

We now build upon the ideas from Section 3 to derive a spectrum of off-policy estimators that allows for interpolation between the trajectory-based PDIS and distribution-based SIS estimators. This spectrum contains PDIS and SIS at the endpoints and allows for smooth interpolation between them to obtain new estimators that can often trade-off the strengths and weaknesses of PDIS and SIS. An illustration of the key idea can be found in Figure 2.

One simple way to perform this trade-off is to control the number of terms in the product in the action likelihood ratio for each reward $R_t$. Specifically, for any reward $R_t$, we propose including only the $n$ most recent action likelihood ratios $\boldsymbol{\rho}_{t-n+1:t}$ in the importance weight, rather than $\boldsymbol{\rho}_{1:t}$. Thus, the overall importance weight becomes the re-weighted probability of visiting $(S_{t-n}, A_{t-n})$, followed by the re-weighted probability of taking the last $n$ actions leading up to reward $R_t$. This reduces the exponential impact that horizon length $L$ has on the variance of PDIS, and provides control over this reduction via the parameter $n$. To get an estimator to perform this trade-off, we start with the derivation in (1) with $z(t) = t - n$, then accumulate the time-dependent state-action distributions $d_t$ over time. The final expression for the finite horizon setting requires some additional constructs and is thus presented along with its derivations and additional discussion in Appendix B. In the following we present the result for the infinite horizon setting.

$$J(\pi_e) = \mathbf{E}_{\boldsymbol{\tau} \sim p_{\pi_b}} \left[ \sum_{t=1}^{n} \gamma^{t-1} \rho_{1:t} R_t + \sum_{t=n+1}^{\infty} \gamma^{t-1} \frac{d^{\pi_e}(S_{t-n}, A_{t-n})}{d^{\pi_b}(S_{t-n}, A_{t-n})} \rho_{t-n+1:t} R_t \right]. \tag{2}$$

Using the sample estimate of (2), we obtain the Spectrum of Off-Policy Estimators (SOPE$_n$),

$$\text{SOPE}_n(D) = \frac{1}{m} \sum_{i=1}^{m} \left( \sum_{t=1}^{n} \gamma^{t-1} \rho_{1:t}^i R_t^i + \sum_{t=n+1}^{\infty} \gamma^{t-1} \hat{w}(S_{t-n}^i, A_{t-n}^i) \rho_{t-n+1:t}^i R_t^i \right).$$

**Remark 1.** *Note that since we generally do not have access to the true density ratios, in practice we substitute $w$ with the estimated density ratios $\hat{w}$ similarly as in SIS. Since SOPE$_n$ is agnostic to how $\hat{w}$ is estimated, it can readily leverage existing and new methods for estimating $\hat{w}$.*

Observe that SOPE$_n$ doesn't just give a single estimator, but a spectrum of off-policy estimators indexed by $n$. An illustration of this spectrum can be seen in Figure 3. As $n$ decreases, the number of terms in the action likelihood ratio decreases, and SOPE$_n$ depends more on the distribution correction ratio and is more like SIS. Likewise as $n$ increases, the number of terms in the action likelihood ratio increases, and SOPE$_n$ is closer to PDIS. Further note that that for the endpoint values of this

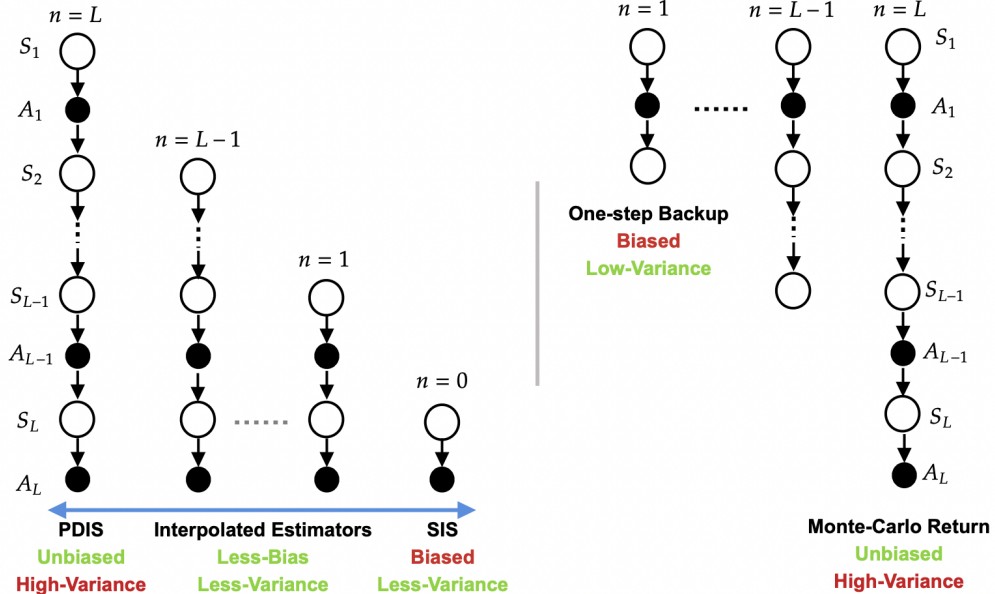

Figure 3: On the left side of the figure, we show an illustration the $\text{SOPE}_n$ spectrum of estimators. For the purpose of this illustration, consider that only at the last time step there is a non-zero reward $R_L$. The $\text{SOPE}_n$ spectrum allows for control of how much an estimate depends on distribution ratios vs action likelihood ratios. Notice that $\text{SOPE}_0$ results in SIS, $\text{SOPE}_L$ results in PDIS estimator, and other values of $n$ result in new interpolated estimators. As an analogy, consider the backup-diagram [Sutton and Barto, 2018] for the n-step q-estimate as illustrated on the right-hand side of the solid vertical line. Notice that in the $n$-step q-estimate, returns are backed up from possible *future outcomes*, whereas in the $n$-step interpolation estimators the probabilities are 'backed-up' from the possible *histories*. (In the diagram, bias-variance characterization of PDIS and SIS is based on typical practical observations [Voloshin et al., 2019, Fu et al., 2021], however it is worth noting that SIS is not biased when oracle density ratios are available, and there are also edge cases, particularly for short horizon problems, where SIS can have higher variance than PDIS [Liu et al., 2020, Metelli et al., 2020]).

spectrum, $n = 0$ and $n = L$, $\text{SOPE}_n$ gives the SIS and PDIS estimators exactly (for PDIS, horizon length needs to be $L$ instead of $\infty$ for the estimator to be well defined),

$$\text{SOPE}_0(D) = \frac{1}{m} \sum_{i=1}^{m} \sum_{t=1}^{L} \gamma^{t-1} w(S_t^i, A_t^i) R_t^i = \text{SIS}(D),$$

$$\text{SOPE}_L(D) = \frac{1}{m} \sum_{i=1}^{m} \sum_{t=1}^{L} \gamma^{t-1} \rho_{1:t}^i R_t^i = \text{PDIS}(D).$$

## 5 Doubly-Robust and Weighted IS Extensions to $\text{SOPE}_n$

An additional advantage of $\text{SOPE}_n$ is that it can be readily extended to obtain a spectrum for other estimators. For instance, to mitigate variance further a popular technique is to leverage domain knowledge from (imperfect) models using doubly-robust estimators [Jiang and Li, 2016, Jiang and Huang, 2020]. In the following we can create a doubly robust version of the $\text{SOPE}_n$ estimator.

Before moving further, we introduce some additional notation. Let,

$$w(t, n) := \begin{cases} \frac{d^{\pi_e}(S_{t-n}, A_{t-n})}{d^{\pi_b}(S_{t-n}, A_{t-n})} \left( \prod_{j=0}^{n-1} \frac{\pi_e(A_{t-j}|S_{t-j})}{\pi_b(A_{t-j}|S_{t-j})} \right) & \text{if } t > n \\ \prod_{j=1}^{t} \frac{\pi_e(A_j|S_j)}{\pi_b(A_j|S_j)} & 1 \le t \le n \\ 1 & \text{otherwise} \end{cases}$$

Let $q$ be an estimate for the q-value function for $\pi_e$, computed using the (imperfect) model. For brevity, we make the random variable $\boldsymbol{\tau} \sim p_{\pi_b}$ implicit for the expectations in this section. For a given value of $n$, performance (2) of $\pi_e$ can then be expressed as,

$$J(\pi_e) = \mathbf{E}\left[\sum_{t=1}^{\infty} w(t,n)\gamma^{t-1}R_t\right].$$

We now use this form to create a spectrum of doubly-robust estimators,

$$J(\pi_e) = \mathbf{E}\left[\sum_{t=1}^{\infty} w(t,n)\gamma^{t-1}R_t\right] + \underbrace{\mathbf{E}\left[\sum_{t=1}^{\infty} w(t,n)\gamma^{t-1}q(S_t,A_t)\right] - \mathbf{E}\left[\sum_{t=1}^{\infty} w(t,n)\gamma^{t-1}q(S_t,A_t)\right]}_{=0}$$

$$\stackrel{(a)}{=} \mathbf{E}\left[\sum_{t=1}^{\infty} w(t,n)\gamma^{t-1}R_t\right] + \mathbf{E}\left[\sum_{t=1}^{\infty} w(t-1,n)\gamma^{t-1}q(S_t,A_t^{\pi_e})\right] - \mathbf{E}\left[\sum_{t=1}^{\infty} w(t,n)\gamma^{t-1}q(S_t,A_t)\right]$$

$$= \mathbf{E}\left[w(0,n)\gamma^0 q(S_1,A_1^{\pi_e})\right] + \mathbf{E}\left[\sum_{t=1}^{\infty} w(t,n)\gamma^{t-1}\Big(R_t + \gamma q(S_{t+1},A_{t+1}^{\pi_e}) - q(S_t,A_t)\Big)\right]$$

$$= \mathbf{E}\Big[q(S_1,A_1^{\pi_e})\Big] + \mathbf{E}\left[\sum_{t=1}^{\infty} w(t,n)\gamma^{t-1}\Big(R_t + \gamma q(S_{t+1},A_{t+1}^{\pi_e}) - q(S_t,A_t)\Big)\right], \qquad (3)$$

where in (a) we used the notation $A_t^{\pi_e}$ to indicate the $A_t \sim \pi_e(\cdot|S_t)$. Using $A_t^{\pi_e}$ eliminates the need for correcting $A_t$ sampled under $\pi_b$. We define DR-SOPE$_n(D)$ to be the sample estimate of (3), i.e., a doubly-robust form for the SOPE$_n(D)$ estimator. It can now be observed that existing doubly-robust estimators are end-points of DR-SOPE$_n(D)$ (for trajectory-wise settings, horizon length needs to be $L$ instead of $\infty$ for the estimator to be well defined),

DR-SOPE$_L(D)$ = Trajectory-wise DR [Jiang and Li, 2016, Thomas and Brunskill, 2016],
DR-SOPE$_0(D)$ = State-action distribution DR [Jiang and Huang, 2020, Kallus and Uehara, 2020].

A variation of PDIS that can often also help in mitigating the variance of PDIS method is the Consistent Weighted Per-Decision Importance Sampling estimator (CWPDIS) [Thomas, 2015]. CWDPIS renormalizes the importance ratio at each time with the sum of importance weights, which causes CWPDIS to be biased (but consistent) and often have lower variance than PDIS.

$$\text{CWPDIS}(D) := \sum_{t=1}^{L} \gamma^{t-1}\frac{\sum_{i=1}^{m}\rho_{1:t}^i R_t^i}{\sum_{i=1}^{m}\rho_{1:t}^i}.$$

Similar DR-SOPE$_n$, we can create a weighted version of SOPE$_n$ estimator that interpolates between a weighted-version of SIS and CWPDIS:

$$\text{W-SOPE}_n(D) := \sum_{t=1}^{n}\left(\gamma^{t-1}\sum_{i=1}^{m}\frac{\rho_{1:t}^i}{\sum_{i=1}^{m}\rho_{1:t}^i}R_t^i\right) + \sum_{t=n+1}^{\infty}\left(\gamma^{t-1}\sum_{i=1}^{m}\frac{w(S_{t-n}^i,A_{t-n}^i)\rho_{t-n+1:t}^i}{\sum_{i=1}^{m}w(S_{t-n}^i,A_{t-n}^i)\rho_{t-n+1:t}^i}R_t^i\right).$$

Since, unlike PDIS, CWPDIS is a biased (but consistent) estimator, W-SOPE$_n$ interpolates between two biased estimators as endpoints. Nonetheless, we show experimentally in Section 6 that in practice W-SOPE$_n$ estimators for intermediate values of $n$ can still outperform weighted-SIS and CWPDIS.

## 6   Experimental Results

In this section, we present experimental results showing that interpolated estimators within the SOPE$_n$ and W-SOPE$_n$ spectrums can outperform the SIS/weighted-SIS and PDIS/CWPDIS endpoints. In each experiment, we evaluate SOPE$_n$ and W-SOPE$_n$ for different values of $n$ ranging from 0 to $L$. This allows us to compare the different estimators we get for each $n$ and see trends of how the

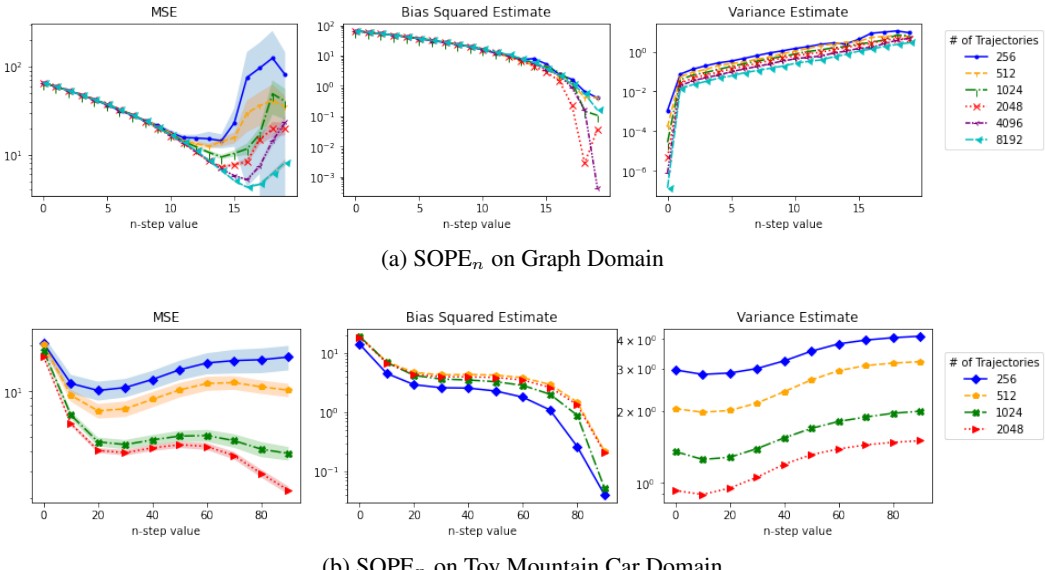

(a) $\text{SOPE}_n$ on Graph Domain

(b) $\text{SOPE}_n$ on Toy Mountain Car Domain

Figure 4: Experimental results from evaluating the $\text{SOPE}_n$ estimator on the Graph and Toy Mountain Car domains. The $x$-axis for each plot indicates the value of $n$ in the $\text{SOPE}_n$ estimate. The shaded regions denote 95% confidence regions on the mean of MSE. Recall that $\text{SOPE}_0$ gives SIS and $\text{SOPE}_L$ gives PDIS. The evaluation and behavior policies are $\pi_e(a = 0) = 0.9$ and $\pi_b(a = 0) = 0.5$ for the experiments on the Graph Domain and and $\pi_e(a = 0) = 0.5$ and $\pi_b(a = 0) = 0.6$ for the Toy Mountain Car domain. In both these domains, we can see that there exist interpolating estimators in the $\text{SOPE}_n$ spectrum that outperform SIS and PDIS, and that the $\text{SOPE}_n$ spectrum empirically performs a bias-variance trade-off.

performance changes as $n$ varies. Additionally, we plot estimates of the bias and the variance for the different values of $n$ to further investigate the properties of estimators in this spectrum.

For our experiments, we utilize the environments and implementations of baseline estimators in the Caltech OPE Benchmarking Suite (COBS) [Voloshin et al., 2019]. In this section, we present results on the Graph and Toy Mountain Car environments. To obtain an estimate of the density ratios $\hat{w}$, we use COBS's implementation of infinite horizon methods from [Liu et al., 2018]. Full experimental details and additional experimental results can be found in Appendix D. Additional experiments include an investigation on the impact on the degree of $\pi_e$ and $\pi_b$ mismatch on $\text{SOPE}_n$ and W-$\text{SOPE}_n$, as well as additional experiments on the Mountain Car domain.

The experimental results for the $\text{SOPE}_n$ and W-$\text{SOPE}_n$ estimators can be seen in Figures 4 and 5 respectively. We observe that for both $\text{SOPE}_n$ and W-$\text{SOPE}_n$, the plots of mean-squared error (MSE) have a U-shape indicating that there exist interpolated estimators within the spectrum with lower MSE than the endpoints. Additionally, from the bias and variance plots, we can see that $\text{SOPE}_n$ performs a bias-variance trade-off in these experiments. We observe that as $n$ increases and the estimators become closer to PDIS, the bias decreases but the variance increases. Likewise, as $n$ decreases and the estimators become closer to SIS, the variance decreases but the bias increases. This bias-variance trade-off trend is very notable for the unweighted $\text{SOPE}_n$ which trades-off between biased SIS and unbiased PDIS endpoints. However, we still can see this trend even with the W-$\text{SOPE}_n$ estimator, although the trade-off is not as clean because W-SOPE interpolates between biased SIS and the also biased (but consistent) CWPDIS.

Finally, note that our plots also show the results for different batch sizes of historical data. In our plots, as batch size increases, for some domains the PDIS/CWPDIS endpoints eventually outperform the SIS/weighted-SIS endpoints. However, even in this case, there still exist interpolated estimators that outperform both endpoints.

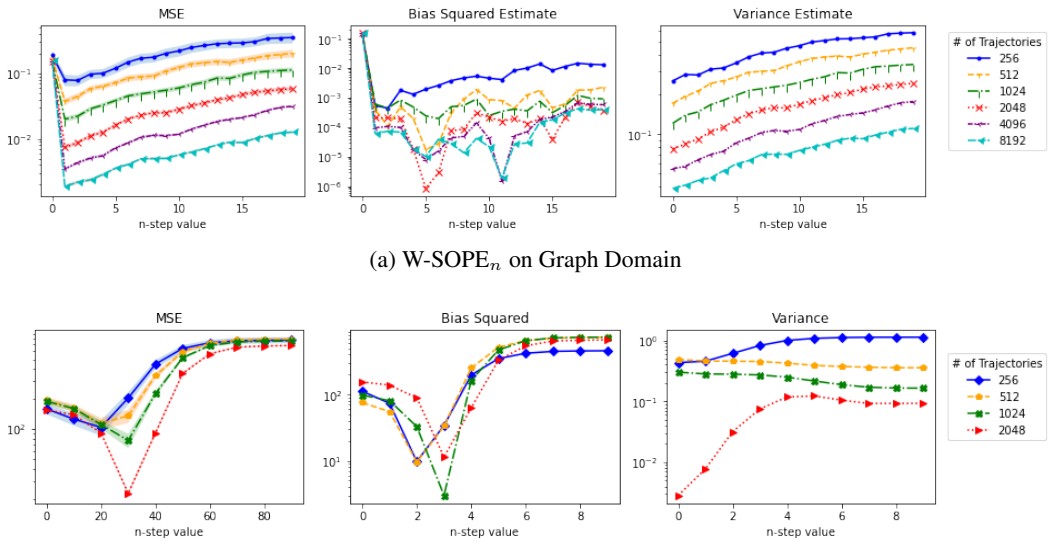

(a) W-SOPE$_n$ on Graph Domain

(b) W-SOPE$_n$ on Toy Mountain Car Domain

Figure 5: Experimental results from evaluating the W-SOPE$_n$ estimator on the Graph and Toy Mountain Car domains. The $x$-axis for each plot indicates the value of $n$ in the SOPE$_n$ estimate. The shaded regions denote 95% confidence regions on the mean of MSE. Recall that W-SOPE$_0$ gives weighted-SIS and W-SOPE$_L$ gives CWPDIS. The evaluation and behavior policies are $\pi_e(a = 0) = 0.9$ and $\pi_b(a = 0) = 0.7$ for the experiments on the Graph Domain and and $\pi_e(a = 0) = 0.5$ and $\pi_b(a = 0) = 0.9$ for the Toy Mountain Car domain. In both these domains, we can see that although we do not get as clean of a bias-variance trade-off as when we use SOPE$_n$, there still exist interpolating estimators in the W-SOPE$_n$ spectrum that outperform SIS and PDIS.

## 7   Related Work

Off-policy evaluation (also related to counterfactual inference in the causality literature [Pearl, 2009]) is one the most crucial aspects of RL, and importance sampling [Metropolis and Ulam, 1949, Horvitz and Thompson, 1952] plays a central role in it. Precup [2000] first introduced IS, PDIS, and WIS estimates for OPE. Since then there has been a flurry of research in this direction: using partial-models to develop doubly robust estimators [Jiang and Li, 2016, Thomas and Brunskill, 2016], using multi-importance sampling [Papini et al., 2019, Metelli et al., 2020], estimating the behavior policy [Hanna et al., 2019], clipping importance ratios [Bottou et al., 2013, Thomas et al., 2015, Munos et al., 2016, Schulman et al., 2017], dropping importance ratios [Guo et al., 2017], importance sampling the entire return distribution [Chandak et al., 2021], importance resampling of trajectories [Schlegel et al., 2019], emphatic weighting of TD methods [Mahmood et al., 2015, Hallak et al., 2016, Patterson et al., 2021], and estimating state-action distributions [Hallak and Mannor, 2017, Liu et al., 2018, Gelada and Bellemare, 2019, Xie et al., 2019, Nachum and Dai, 2020, Yang et al., 2020, Zhang et al., 2020, Jiang and Huang, 2020, Uehara et al., 2020].

Perhaps the most relevant to our work are the recent works by Liu et al. [2020] and Rowland et al. [2020] that use the conditional IS (CIS) framework to show how IS, PDIS, and SIS are special instances of CIS. Similarly, our proposed method for combining trajectory and density-based importance sampling also falls under the CIS framework. Liu et al. [2020] also showed that in the finite horizon setting, none of IS, PDIS, or SIS has variance *always* lesser than the other. Similarly, Rowland et al. [2020] used sufficient conditional functions to create new off-policy estimators and showed that return conditioned estimates (RCIS) can provide optimal variance reduction. However, using RCIS requires a challenging task of estimating *density ratios for returns* (not state-action pair) and Liu et al. [2020] established a negative result that estimating these ratios using linear regression may result in the IS estimate itself.

Our analysis complements these recent works by showing that there exists interpolated estimators that can provide lower variance estimates than any of IS, PDIS, or SIS. Our proposed estimator SOPE$_n$ provides a natural interpolation technique to trade-off between the strengths and weaknesses of these

trajectory and density based methods. Additionally, while it is known that $q^\pi(s, a)$ and $d^\pi(s, a)$ have a primal-dual connection [Wang et al., 2007], our time-based interpolation technique also sheds new light on connections between their n-step generalizations.

# 8 Conclusions

We present a new perspective in off-policy evaluation connecting two popular estimators, PDIS and SIS, and show that PDIS and SIS lie as endpoints on the Spectrum of Off-Policy Estimators $\text{SOPE}_n$ which interpolates between them. Additionally, we also derive a weighted and doubly robust version of this spectrum of estimators. With our experimental results, we illustrate that estimators that lie on the interior of the $\text{SOPE}_n$ and $\text{W-SOPE}_n$ spectrums can be used outperform their endpoints SIS/weighted-SIS and PDIS/CWPDIS.

While we are able to show there exist $\text{SOPE}_n$ estimators that are able to outperform PDIS and SIS, it remains as future work to devise strategies to automatically select $n$ to trade-off bias and variance. Future directions may include developing methods to select $n$ or combine all estimators for all $n$ using $\lambda$-trace methods [Sutton and Barto, 2018] to best trade-off bias and variance.

Finally, like all off-policy evaluation methods, our approach carries risks if used inappropriately. When using OPE for sensitive or safety-critical applications such as medical domains, caution should be taken to carefully consider the variance and bias of the estimator that is used. In these cases, high-confidence OPE methods [Thomas et al., 2015] may be more appropriate.

# 9 Acknowledgement

We thank members of the Personal Autonomous Robotics Lab (PeARL) at the University of Texas at Austin for discussion and feedback on early stages of this work. We especially thank Jordan Schneider, Harshit Sikchi, and Prasoon Goyal for reading and giving suggestions on early drafts. We additionally thank Ziyang Tang for suggesting an additional marginalization step in the main proof that helped us unify the results for finite and infinite horizon setting. The work also benefited from feedback by Nan Jiang during initial stages of this work. We would also like to thank the anonymous reviewers for their suggestions which helped improve the paper.

This work has taken place in part in the Personal Autonomous Robotics Lab (PeARL) at The University of Texas at Austin. PeARL research is supported in part by the NSF (IIS-1724157, IIS-1638107, IIS-1749204, IIS-1925082), ONR (N00014-18-2243), AFOSR (FA9550-20-1-0077), and ARO (78372-CS). This research was also sponsored by the Army Research Office under Cooperative Agreement Number W911NF-19-2-0333, a gift from Adobe, NSF award #2018372, and the DEVCOM Army Research Laboratory under Cooperative Agreement W911NF-17-2-0196 (ARL IoBT CRA). The views and conclusions contained in this document are those of the authors and should not be interpreted as representing the official policies, either expressed or implied, of the Army Research Office or the U.S. Government. The U.S. Government is authorized to reproduce and distribute reprints for Government purposes notwithstanding any copyright notation herein.

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
