# A  Proof of Lemma 1

Liu et al. [2018] first showed that stationary importance sampling methods can be viewed as Rao-Blackwellization of IS estimator, and claimed that the expectation of the likelihood-ratios conditioned on state and action is equal to the distribution ratio, as stated in Property 1. For completeness, we present a proof of Property 1. Recall that $d_t^\pi(s,a) = p_\pi(S_t = s, A_t = a)$.

$$\mathbf{E}_{\boldsymbol{\tau} \sim p_{\pi_b}} \left[ \rho_{1:t} | S_t = s, A_t = a \right]$$

$$= \mathbf{E}_{\boldsymbol{\tau} \sim p_{\pi_b}} \left[ \frac{p_{\pi_e}(\boldsymbol{\tau}_{1:t})}{p_{\pi_b}(\boldsymbol{\tau}_{1:t})} \bigg| S_t = s, A_t = a \right]$$

$$= \mathbf{E}_{\boldsymbol{\tau} \sim p_{\pi_b}} \left[ \frac{p_{\pi_e}(S_1, A_1, \ldots, S_t, A_t)}{p_{\pi_b}(S_1, A_1, \ldots, S_t, A_t)} \bigg| S_t = s, A_t = a \right]$$

$$= \mathbf{E}_{\boldsymbol{\tau} \sim p_{\pi_b}} \left[ \frac{p_{\pi_e}(S_1, A_1, \ldots, S_t, A_t)}{p_{\pi_e}(S_t, A_t)} \frac{p_{\pi_b}(S_t, A_t)}{p_{\pi_b}(S_1, A_1, \ldots, S_t, A_t)} \frac{p_{\pi_e}(S_t, A_t)}{p_{\pi_b}(S_t, A_t)} \bigg| S_t = s, A_t = a \right]$$

$$= \mathbf{E}_{\boldsymbol{\tau} \sim p_{\pi_b}} \left[ \frac{p_{\pi_e}(\boldsymbol{\tau}_{1:t}|S_t, A_t)}{p_{\pi_b}(\boldsymbol{\tau}_{1:t}|S_t, A_t)} \bigg| S_t = s, A_t = a \right] \frac{p_{\pi_e}(S_t = s, A_t = a)}{p_{\pi_b}(S_t = s, A_t = a)}$$

$$\stackrel{(a)}{=} \mathbf{E}_{\boldsymbol{\tau} \sim p_{\pi_b}} \left[ \frac{p_{\pi_e}(\boldsymbol{\tau}_{1:t}|S_t, A_t)}{p_{\pi_b}(\boldsymbol{\tau}_{1:t}|S_t, A_t)} \bigg| S_t = s, A_t = a \right] \frac{d_t^{\pi_e}(s,a)}{d_t^{\pi_b}(s,a)}$$

$$= \left( \sum_\tau \frac{p_{\pi_e}(\tau_{1:t}|S_t = s, A_t = a)}{p_{\pi_b}(\tau_{1:t}|S_t = s, A_t = a)} p_{\pi_b}(\tau|S_t = s, A_t = a) \right) \frac{d_t^{\pi_e}(s,a)}{d_t^{\pi_b}(s,a)}$$

$$\stackrel{(b)}{=} \left( \sum_\tau \frac{p_{\pi_e}(\tau_{1:t}|S_t = s, A_t = a)}{p_{\pi_b}(\tau_{1:t}|S_t = s, A_t = a)} p_{\pi_b}(\tau_{1:t}|S_t = s, A_t = a) p_{\pi_b}(\tau_{t+1:L}|S_t = s, A_t = a) \right) \frac{d_t^{\pi_e}(s,a)}{d_t^{\pi_b}(s,a)}$$

$$\stackrel{(c)}{=} \left( \sum_{\tau_{1:t}} p_{\pi_e}(\tau_{1:t}|S_t = s, A_t = a) \sum_{\tau_{t+1:L}} p_{\pi_b}(\tau_{t+1:L}|S_t = s, A_t = a) \right) \frac{d_t^{\pi_e}(s,a)}{d_t^{\pi_b}(s,a)}$$

$$= \frac{d_t^{\pi_e}(s,a)}{d_t^{\pi_b}(s,a)}.$$

Line (a) follows from $d_t^\pi(s,a) = p_\pi(S_t = s, A_t = a)$. In line (b), we use the Markov property which gives that $\tau_{1:t}$ and $\tau_{t+1:L}$ are independent conditioned on $(S_t = s, A_t = a)$. Line (c) follows from splitting the summation over $\tau$ into to summations over $\tau_{1:t}$ and $\tau_{t+1:L}$.

# B  Full Derivation of SOPE$_n$ Estimator

To derive the SOPE$_n$ estimator, we repeat the derivation of (1) with $z$ being a function of time, $z(t) = \max\{t - n, 0\}$. This gives us the expression

$$J(\pi_e) = \mathbf{E}_{\boldsymbol{\tau} \sim p_{\pi_b}} \left[ \sum_{t=1}^n \gamma^{t-1} \rho_{1:t} R_t + \sum_{t=n+1}^L \gamma^{t-1} \frac{d_{t-n}^{\pi_e}(S_{t-n}, A_{t-n})}{d_{t-n}^{\pi_b}(S_{t-n}, A_{t-n})} \rho_{t-n+1:t} R_t \right]. \quad (4)$$

Since $z(t)$ is function of $t$, we can accumulate the $d_t^\pi$ across time so that we can write the interpolating expression using *average* state-action distribution ratios, rather than time-dependent ones. This additional marginalization step over time allows us to consider time-independent distribution ratios. Notation-wise, let $d_{1:T}^\pi := (\sum_{t=1}^T \gamma^{t-1} d_t^\pi(s,a))/(\sum_{t=1}^T \gamma^{t-1})$ for any time $T$. $d_{1:T}$ can be thought of as at the average state-action visitation over the first $T$ time-steps. Note that $d^\pi = \lim_{T \to \infty} d_{1:T}^\pi$ where $d^\pi$ is the average state-action distribution. Then, using the law of total expectation, we can write the expectation of the second sum in (4) as:

$$\mathbf{E}_{\boldsymbol{\tau} \sim p_{\pi_b}} \left[ \sum_{t=n+1}^L \gamma^{t-1} \frac{d_{t-n}^{\pi_e}(S_{t-n}, A_{t-n})}{d_{t-n}^{\pi_b}(S_{t-n}, A_{t-n})} \rho_{t-n+1:t} R_t \right]$$

$$= \sum_{t=n+1}^L \gamma^{t-1} \mathbf{E}_{\substack{(S_{t-n}, A_{t-n}) \\ \sim d_{t-n}^{\pi_b}}} \left[ \mathbf{E}_{\boldsymbol{\tau} \sim p_{\pi_b}} \left[ \frac{d_{t-n}^{\pi_e}(S_{t-n}, A_{t-n})}{d_{t-n}^{\pi_b}(S_{t-n}, A_{t-n})} \rho_{t-n+1:t} R_t \bigg| S_{t-n}, A_{t-n} \right] \right]$$

$$= \sum_{t=n+1}^{L} \gamma^{t-1} \mathbf{E}_{\substack{(S_{t-n}, A_{t-n}) \\ \sim d_{t-n}^{\pi_b}}} \left[ \frac{d_{t-n}^{\pi_e}(S_{t-n}, A_{t-n})}{d_{t-n}^{\pi_b}(S_{t-n}, A_{t-n})} \mathbf{E}_{\tau \sim p_{\pi_b}} \left[ \rho_{t-n+1:t} R_t | S_{t-n}, A_{t-n} \right] \right]$$

$$= \sum_{t=n+1}^{L} \gamma^{t-1} \sum_{s,a} d_{t-n}^{\pi_b}(s,a) \frac{d_{t-n}^{\pi_e}(s,a)}{d_{t-n}^{\pi_b}(s,a)} \mathbf{E}_{\tau \sim p_{\pi_b}} \left[ \rho_{t-n+1:t} R_t | S_{t-n} = s, A_{t-n} = a \right]$$

$$= \sum_{t=n+1}^{L} \gamma^{t-1} \sum_{s,a} d_{t-n}^{\pi_e}(s,a) \mathbf{E}_{\tau \sim p_{\pi_b}} \left[ \rho_{t-n+1:t} R_t | S_{t-n} = s, A_{t-n} = a \right]$$

$$\overset{(a)}{=} \sum_{s,a} \left( \sum_{t=n+1}^{L} \gamma^{t-1} d_{t-n}^{\pi_e}(s,a) \right) \mathbf{E}_{\tau \sim p_{\pi_b}} \left[ \rho_{1:n} R_n | S_1 = s, A_1 = a \right]$$

$$= \sum_{s,a} \left( \sum_{t=1}^{L-n} \gamma^{t-1} d_{t}^{\pi_e}(s,a) \right) \mathbf{E}_{\tau \sim p_{\pi_b}} \left[ \rho_{1:n} R_n | S_1 = s, A_1 = a \right]$$

$$\overset{(b)}{=} \sum_{s,a} \left( \sum_{t=1}^{L-n} \gamma^{t-1} \right) d_{1:L-n}^{\pi_e}(s,a) \mathbf{E}_{\tau \sim p_{\pi_b}} \left[ \rho_{1:n} R_n | S_1 = s, A_1 = a \right]$$

$$\overset{(c)}{=} \sum_{s,a} \left( \sum_{t=1}^{L-n} \gamma^{t-1} \right) d_{1:L-n}^{\pi_b}(s,a) \frac{d_{1:L-n}^{\pi_e}(s,a)}{d_{1:L-n}^{\pi_b}(s,a)} \mathbf{E}_{\tau \sim p_{\pi_b}} \left[ \rho_{1:n} R_n | S_1 = s, A_1 = a \right]$$

$$\overset{(d)}{=} \sum_{s,a} \left( \sum_{t=1}^{L-n} \gamma^{t-1} d_{t}^{\pi_b}(s,a) \right) \frac{d_{1:L-n}^{\pi_e}(s,a)}{d_{1:L-n}^{\pi_b}(s,a)} \mathbf{E}_{\tau \sim p_{\pi_b}} \left[ \rho_{1:n} R_n | S_1 = s, A_1 = a \right]$$

$$= \sum_{s,a} \left( \sum_{t=n+1}^{L} \gamma^{t-1} d_{t-n}^{\pi_b}(s,a) \right) \frac{d_{1:L-n}^{\pi_e}(s,a)}{d_{1:L-n}^{\pi_b}(s,a)} \mathbf{E}_{\tau \sim p_{\pi_b}} \left[ \rho_{1:n} R_n | S_1 = s, A_1 = a \right]$$

$$= \sum_{t=n+1}^{L} \gamma^{t-1} \sum_{s,a} d_{t-n}^{\pi_b}(s,a) \frac{d_{1:L-n}^{\pi_e}(s,a)}{d_{1:L-n}^{\pi_b}(s,a)} \mathbf{E}_{\tau \sim p_{\pi_b}} \left[ \rho_{t-n+1:t} R_t | S_{t-n} = s, A_{t-n} = a \right]$$

$$= \sum_{t=n+1}^{L} \gamma^{t-1} \mathbf{E}_{\substack{(S_{t-n}, A_{t-n}) \\ \sim d_{t-n}^{\pi_b}}} \left[ \mathbf{E}_{\tau \sim p_{\pi_b}} \left[ \frac{d_{1:L-n}^{\pi_e}(S_{t-n}, A_{t-n})}{d_{1:L-n}^{\pi_b}(S_{t-n}, A_{t-n})} \rho_{t-n+1:t} R_t \middle| S_{t-n}, A_{t-n} \right] \right]$$

$$= \mathbf{E}_{\tau \sim p_{\pi_b}} \left[ \sum_{t=n+1}^{L} \gamma^{t-1} \frac{d_{1:L-n}^{\pi_e}(S_{t-n}, A_{t-n})}{d_{1:L-n}^{\pi_b}(S_{t-n}, A_{t-n})} \rho_{t-n+1:t} R_t \right]. \tag{5}$$

In line (a), we use $\mathbf{E}_{\tau \sim p_{\pi_b}} [\rho_{t-n+1:t} R_t | S_{t-n} = s, A_{t-n} = a] = \mathbf{E}_{\tau \sim p_{\pi_b}} [\rho_{1:n} R_n | S_1 = s, A_1 = a]$ which follows from noting that conditioning on $S_{t-n}, A_{t-n}$ and considering the $n$ time steps after is equivalent to conditioning on $S_1, A_1$ and considering the $n$ time steps after that. Lines (b) and (d) follow from $d_{1:L-n}^{\pi} = \left( \sum_{t=1}^{L-n} \gamma^{t-1} d_t^{\pi}(s,a) \right) / \left( \sum_{t=1}^{L-n} \gamma^{t-1} \right)$. Line (c) is possible due to Assumption 1. Plugging in the final expression from (5) back into (4) gives us

$$J(\pi_e) = \mathbf{E}_{\tau \sim p_{\pi_b}} \left[ \sum_{t=1}^{n} \gamma^{t-1} \rho_{1:t} R_t + \sum_{t=n+1}^{L} \gamma^{t-1} \frac{d_{1:L-n}^{\pi_e}(S_{t-n}, A_{t-n})}{d_{1:L-n}^{\pi_b}(S_{t-n}, A_{t-n})} \rho_{t-n+1:t} R_t \right]. \tag{6}$$

Note that $\frac{d_{1:L-n}^{\pi_e}(s,a)}{d_{1:L-n}^{\pi_b}(s,a)}$ is the state-action distribution ratio over the first $L-n$ time-steps. In practice, to estimate this ratio, one can discard the data from time-step $L-n$ to $L$, and use the same min-max optimization procedures used to estimate $\frac{d_{1:L}^{\pi_e}(s,a)}{d_{1:L}^{\pi_b}(s,a)}$ on the remaining data to estimate this ratio.

Note that in the infinite horizon setting where $L \to \infty$ and for finite $n$, (6) becomes

$$J(\pi_e) = \mathbf{E}_{\tau \sim p_{\pi_b}} \left[ \sum_{t=1}^{n} \gamma^{t-1} \rho_{1:t} R_t + \sum_{t=n+1}^{\infty} \gamma^{t-1} \frac{d^{\pi_e}(S_{t-n}, A_{t-n})}{d^{\pi_b}(S_{t-n}, A_{t-n})} \rho_{t-n+1:t} R_t \right].$$

In this case, the typical optimization procedures for estimating $\frac{d^{\pi_e}(s,a)}{d^{\pi_b}(s,a)}$ in the infinite horizon setting can be used to estimate the distribution ratios.

Additionally, note that specifically for the infinite horizon setting, we can alternatively derive the $\text{SOPE}_n$ estimator using the Bellman equations for the average state-action distribution $d^\pi$. This alternative derivation can be found in Appendix C.

## C  Bellman Recursion Derivation of $\text{SOPE}_n$

We present an alternative derivation of the $\text{SOPE}_n$ estimator for the infinite horizon setting using the Bellman equations for the average state-action distribution $d^\pi$, which is:

$$d^\pi(s,a) := (1-\gamma)\sum_{t=1}^{\infty}\gamma^{t-1}\Pr(S_t = s, A_t = a\,;\pi)$$

$$= (1-\gamma)d_1(s)\pi(a|s) + \gamma\sum_{s'\in\mathcal{S},a'\in\mathcal{A}}\Pr(s,a|s',a'\,;\pi)d^\pi(s',a'). \tag{7}$$

Now using (7) we can expand $J(\pi_e)$ and unroll $d^{\pi_e}$ once to obtain

$$J(\pi_e) = (1-\gamma)^{-1}\sum_{s\in\mathcal{S},a\in\mathcal{A}}r(s,a)d^{\pi_e}(s,a)$$

$$= (1-\gamma)^{-1}\sum_{s\in\mathcal{S},a\in\mathcal{A}}r(s,a)\left[(1-\gamma)d_1(s)\pi_e(a|s) + \gamma\sum_{s'\in\mathcal{S},a'\in\mathcal{A}}\Pr(s,a|s',a'\,;\pi_e)d^{\pi_e}(s',a')\right]$$

$$= \sum_{s\in\mathcal{S},a\in\mathcal{A}}r(s,a)d_1(s)\pi_e(a|s) + \gamma(1-\gamma)^{-1}\sum_{s\in\mathcal{S},a\in\mathcal{A}}\sum_{s'\in\mathcal{S},a'\in\mathcal{A}}\Pr(s,a|s',a'\,;\pi_e)d^{\pi_e}(s',a')r(s,a)$$

$$\overset{(a)}{=} \sum_{s\in\mathcal{S},a\in\mathcal{A}}r(s,a)d_1(s)\pi_e(a|s) + \gamma(1-\gamma)^{-1}\sum_{s\in\mathcal{S},a\in\mathcal{A}}\sum_{s'\in\mathcal{S},a'\in\mathcal{A}}\Pr(s',a'|s,a\,;\pi_e)d^{\pi_e}(s,a)r(s',a')$$

$$= \sum_{s\in\mathcal{S},a\in\mathcal{A}}\pi_b(a|s)r(s,a)d_1(s)\frac{\pi_e(a|s)}{\pi_b(a|s)}$$

$$+ \gamma(1-\gamma)^{-1}\sum_{s\in\mathcal{S},a\in\mathcal{A}}d^{\pi_b}(s,a)\sum_{s'\in\mathcal{S},a'\in\mathcal{A}}\pi_b(a'|s')\Pr(s'|s,a)\frac{\pi_e(a'|s')}{\pi_b(a'|s')}\frac{d^{\pi_e}(s,a)}{d^{\pi_b}(s,a)}r(s',a')$$

$$= \sum_{s\in\mathcal{S},a\in\mathcal{A}}\pi_b(a|s)r(s,a)d_1(s)\frac{\pi_e(a|s)}{\pi_b(a|s)}$$

$$+ \gamma\sum_{s\in\mathcal{S},a\in\mathcal{A}}\sum_{t=1}^{\infty}\gamma^{t-1}\Pr(S_t = s, A_t = a\,;\pi_b)\sum_{s'\in\mathcal{S},a'\in\mathcal{A}}\pi_b(a'|s')\Pr(s'|s,a)\frac{\pi_e(a'|s')}{\pi_b(a'|s')}\frac{d^{\pi_e}(s,a)}{d^{\pi_b}(s,a)}r(s',a')$$

$$= \mathbf{E}_{\boldsymbol{\tau}\sim\pi_b}\left[\frac{\pi_e(A_1|S_1)}{\pi_b(A_1|S_1)}r(S_1,A_1) + \sum_{t=1}^{\infty}\gamma^t\frac{d^{\pi_e}(S_t,A_t)}{d^{\pi_b}(S_t,A_t)}\frac{\pi_e(A_{t+1}|S_{t+1})}{\pi_b(A_{t+1}|S_{t+1})}r(S_{t+1},A_{t+1})\right]$$

$$= \mathbf{E}_{\boldsymbol{\tau}\sim\pi_b}\left[\frac{\pi_e(A_1|S_1)}{\pi_b(A_1|S_1)}r(S_1,A_1) + \sum_{t=2}^{\infty}\gamma^{t-1}\frac{d^\pi(S_{t-1},A_{t-1})}{d^{\pi_b}(S_{t-1},A_{t-1})}\frac{\pi_e(A_t|S_t)}{\pi_b(A_t|S_t)}r(S_t,A_t)\right]. \tag{8}$$

where (a) follows by relabelling in the common notation such that $(s,a)$ and $(s',a')$ are consecutive state-action pairs. Notice that $\text{SOPE}_1(D)$ is the sample estimate of (8). Similarly, on unrolling $d^{\pi_b}$ twice using (7),

$$J(\pi_e) = (1-\gamma)^{-1} \sum_{s\in\mathcal{S},a\in\mathcal{A}} r(s,a)d^{\pi_e}(s,a)$$

$$= (1-\gamma)^{-1} \sum_{s\in\mathcal{S},a\in\mathcal{A}} r(s,a)\Bigg[(1-\gamma)d_1(s)\pi_e(a|s)$$

$$+ \gamma \sum_{s'\in\mathcal{S},a'\in\mathcal{A}} \Pr(s,a|s',a';\pi_e)\bigg[(1-\gamma)d_1(s')\pi_e(a'|s') + \gamma \sum_{s''\in\mathcal{S},a''\in\mathcal{A}} \Pr(s',a'|s'',a'';\pi_e)d^{\pi_e}(s'',a'')\bigg]\Bigg]$$

$$= \sum_{s\in\mathcal{S},a\in\mathcal{A}} r(s,a)d_1(s)\pi_e(a|s) + \gamma \sum_{s\in\mathcal{S},a\in\mathcal{A}} r(s,a) \sum_{s'\in\mathcal{S},a'\in\mathcal{A}} \Pr(s,a|s',a';\pi_e)d_1(s')\pi_e(a'|s')$$

$$+ \gamma^2(1-\gamma)^{-1} \sum_{s\in\mathcal{S},a\in\mathcal{A}} r(s,a) \sum_{s'\in\mathcal{S},a'\in\mathcal{A}} \Pr(s,a|s',a';\pi_e) \sum_{s''\in\mathcal{S},a''\in\mathcal{A}} \Pr(s',a'|s'',a'';\pi_e)d^{\pi_e}(s'',a'')$$

$$= \sum_{s\in\mathcal{S},a\in\mathcal{A}} r(s,a)d_1(s)\pi_e(a|s) + \gamma \sum_{s'\in\mathcal{S},a'\in\mathcal{A}} r(s',a') \sum_{s\in\mathcal{S},a\in\mathcal{A}} \Pr(s',a'|s,a;\pi_e)d_1(s)\pi_e(a|s)$$

$$+ \gamma^2(1-\gamma)^{-1} \sum_{s''\in\mathcal{S},a''\in\mathcal{A}} r(s'',a'') \sum_{s'\in\mathcal{S},a'\in\mathcal{A}} \Pr(s'',a''|s',a';\pi_e) \sum_{s\in\mathcal{S},a\in\mathcal{A}} \Pr(s',a'|s,a;\pi_e)d^{\pi_e}(s,a),$$

Where the last line follows by relabelling the state-action pairs such that they match the common notation where $(s,a)$, $(s',a')$ and $(s'',a'')$ are the state action tuples for three consecutive time-steps. Now changing the sampling distribution as earlier,

$$J(\pi_e) = \mathbf{E}_{\boldsymbol{\tau}\sim\pi_b}\Bigg[\frac{\pi_e(A_1|S_1)}{\pi_b(A_1|S_1)}r(S_1,A_1) + \gamma\frac{\pi_e(A_1|S_1)}{\pi_b(A_1|S_1)}\frac{\pi_e(A_2|S_2)}{\pi_b(A_2|S_2)}r(S_2,A_2)$$

$$+ \sum_{t=3}^{\infty}\gamma^{t-1}\frac{d_e^{\pi}(S_{t-2},A_{t-2})}{d^{\pi_b}(S_{t-2},A_{t-2})}\frac{\pi_e(A_{t-1}|S_{t-1})}{\pi_b(A_{t-1}|S_{t-1})}\frac{\pi_e(A_t|S_t)}{\pi_b(A_t|S_t)}r(S_t,A_t)\Bigg] \qquad (9)$$

It can be now observed that $\text{SOPE}_2(D)$ is the sample estimate of (9). Similarly, by generalizing this pattern it can be observed that on unrolling $n$ times, we will get,

$$J(\pi_e) = \mathbf{E}_{\boldsymbol{\tau}\sim\pi_b}\Bigg[\sum_{t=1}^{n}\left(\prod_{j=1}^{t}\frac{\pi_e(A_j|S_j)}{\pi_b(A_j|S_j)}\right)\gamma^{t-1}r(S_t,A_t)+$$

$$\sum_{t=n+1}^{\infty}\gamma^{t-1}\frac{d^{\pi_e}(S_{t-n},A_{t-n})}{d^{\pi_b}(S_{t-n},A_{t-n})}\left(\prod_{j=0}^{n-1}\frac{\pi_e(A_{t-j}|S_{t-j})}{\pi_b(A_{t-j}|S_{t-j})}\right)r(S_t,A_t)\Bigg]$$

$$= \mathbf{E}_{\boldsymbol{\tau}\sim p_{\pi_b}}\Bigg[\sum_{t=1}^{n}\gamma^{t-1}\rho_{1:t}R_t + \sum_{t=n+1}^{\infty}\gamma^{t-1}\frac{d^{\pi_e}(S_{t-n},A_{t-n})}{d^{\pi_b}(S_{t-n},A_{t-n})}\rho_{t-n+1:t}R_t\Bigg]. \qquad (10)$$

Finally, it can be observed that that $\text{SOPE}_n(D)$ is the sample estimate of (10).

## D   Additional Experimental Details

For all experiments, we utilize the domains and algorithm implementations from Caltech OPE Benchmarking Suite (COBS) library by Voloshin et al. [2019]. Our code can be found at https://github.com/Pearl-UTexas/SOPE, and our experiments ran on 32 Intel Xeon cores.

## D.1 Experimental Set-Up

For our experiments, we used the Graph, Toy Mountain Car, and standard Mountain Car [Brockman et al., 2016] domains provided in the COBS library. We include a brief description of each of these domains below, and a full description of each can be found in the work by Voloshin et al. [2019].

**Graph Environment** The Graph environment is a two-chain environment with $2L$ states and 2 actions. The ends of the chain are starting state $x_0 = 0$ and absorbing state $x_{abs} = 2L$. In between $x_0$ and $x_{abs}$, the remaining states form two chains of length $L - 1$ each. The states on the top chain are labeled $1, 3, \ldots, 2L - 3$ and the states on the bottom chain are labeled $2, 4, \ldots, 2L - 2$. For each $t < L$, taking action $a = 0$, the agent will try to enter the next state on the top chain $x_{t+1} = 2t + 1$, and taking action $a = 1$, the agent will try to enter the next state on the bottom chain $x_{t+1} = 2t + 2$. Since the environment is stochastic, the agent will succeed with probability 0.75 and slip into the wrong row with probability 0.25. The reward is +1 if the agent transitions to a state on the top chain and -1 otherwise. For our experiments, we set $L = 20$ and $\gamma = 0.98$.

**Toy Mountain Car Environment** The Toy-MC environment [Voloshin et al., 2019] is a tabular simplification of the classic Mountain Car domain. There are a total of 21 states: $x_0 = 0$ the starting point in the valley, 10 states to the left, and 10 states to the right. The right-most state is a terminal absorbing state. Taking action $a = 0$ moves the agent to the right and taking action $a = 1$ moves the agent to the left. The agent receives reward of $r = -1$ each time step, and the reward becomes 0 when the agent reaches the terminal absorbing states. For our experiments, we use random restart where start in a random state in the domain and set $L = 100$ and $\gamma = 0.99$.

**Mountain Car Environment** We use the Mountain Car environment from OpenAI gym with the simplifying modifications applied in Voloshin et al. [2019]. In particular, the car agent starts in a valley and needs to move back and forth in order to gain moment to reach the goal of getting to the top of the mountain. The state space is the position and velocity of the car. At each time step, the car agent can either accelerate move forward, move backwards, or do nothing. Additionally, at each time, the agent receives a reward of $r = -1$ until it reaches the goal. The environment is modified in the COBS library to decrease the effective trajectory length by applying each action $a_t$ five times before observing $x_{t+1}$. Additionally, the initial start location is modified from being uniformly chosen between $[-.6, -.4]$ to be randomly chosen from $\{-.6, -.5, -.4\}$ with no velocity.

**Policies** For the tabular environments Graph and Toy Mountain Car, we utilize static policies that take action $a = 0$ with probability $p$ and action $a = 1$ with probability $1 - p$. For the Mountain Car environment, we utilize an $\epsilon$-greedy policies with the provided DDQN trained policy in the COBS library.

**Methods** For our experiments, we evaluate the performance of our proposed SOPE$_n$ and W-SOPE$_n$ estimators. To estimate the average state-action visitation ratios $\frac{d^{\pi_e}(s,a)}{d^{\pi_b}(s,a)}$, we utilize the implementation of methods from Liu et al. [2018] provided in the COBS library. For the Mountain Car experiments, we utilize the radial-basis function for the kernel estimate and a linear function class for the density estimate. Specific hyper-parameters can be found below.

| Parameter | Graph | Toy-MC | Mountain Car |
|---|---|---|---|
| Quad. prog. regular. | 1e-3 | 1e-3 | - |
| NN Fit Epochs | - | - | 1000 |
| NN Batchsize | - | - | 1024 |

## D.2 Impact of Policy Mismatch Between $\pi_b$ and $\pi_e$ on SOPE$_n$ and W-SOPE$_n$

We examine the impact of the policy mismatch between the behavior and evaluation policies on the performance of the SOPE$_n$ and W-SOPE$_n$ estimators. In this experiment, the evaluation policy takes action $a = 0$ with probability 0.9, and we vary the probability that the behavior policy takes $a = 0$ from 0.1 to 0.8 by increments of 0.1. We examine the performance of the SOPE$_n$ and W-SOPE$_n$ estimators across values of $n$ for the different behavior policies. Results can be seen in the plots below.

The performance of PDIS and SIS has been known to be negatively correlated with the degree of policy mismatch [Voloshin et al., 2019]. We also find this to be generally true for the performance of the SOPE$_n$ and W-SOPE$_n$ estimators. Additionally, we observe that the degree of mismatch between

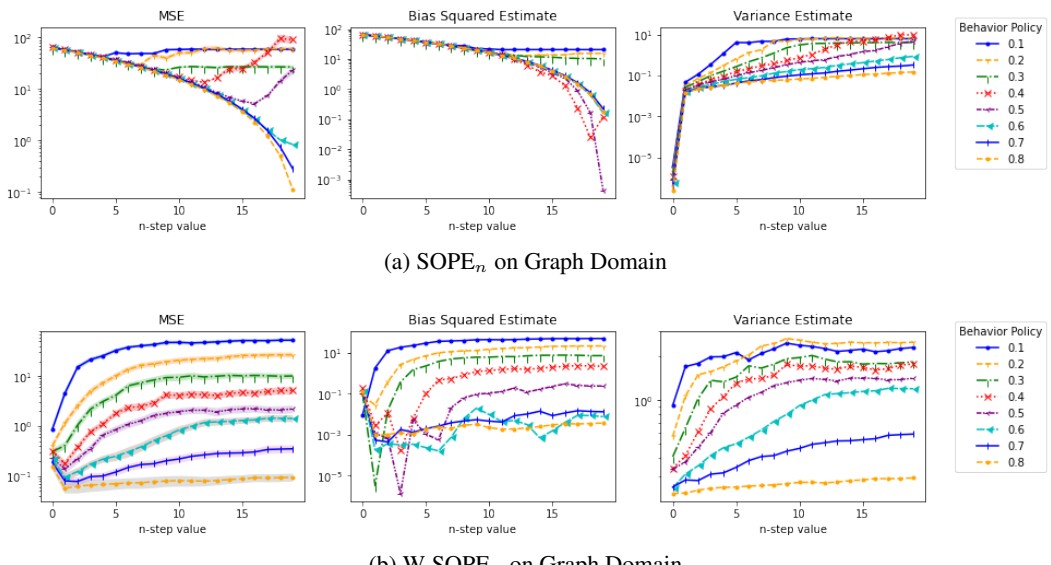

(a) SOPE$_n$ on Graph Domain

(b) W-SOPE$_n$ on Graph Domain

the evaluation and behavior policies has an impact on the existence of an interpolating estimator that is able to achieve lower MSE than the endpoints. For both SOPE$_n$ and W-SOPE$_n$, when the $\pi_b$ is extremely different $\pi_e$, there are instances when the best estimate is SIS or weighted-SIS. In cases when the $\pi_b$ is extremely close to the $\pi_e$, particularly for unweighted SOPE$_n$, there are cases when the trajectory-based importance sampling endpoint gives the lowest MSE. We do note that in cases when the difference between evaluation and behavior policies moderate but not extreme, there exists interpolating estimators that outperform the endpoints. This experiment helps to shed light on the possible conditions on the evaluation and behavior policies that allow for an interpolating estimator to have the lowest MSE.

### D.3 Mountain Car Experimental Results

In addition to the experiments contained in the main paper, we also examine the performance of W-SOPE$_n$ on the Mountain Car domain. For these experiments, we used a provided DDQN trained policy as the base policy, and use $\epsilon$-greedy versions of this policy as our behavior and evaluation policies. Specific information about this policy can be found in [Voloshin et al., 2019]. For our behavior policy, we use $\epsilon = 0.05$ and for our evaluation policy, we use $\epsilon = 0.9$. We average over 10 trials with $128, 256$ and $512$ trajectories each. The results of this experiment can be found in the figure below.

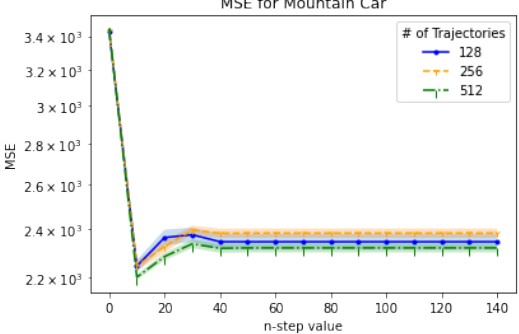

We observe that this setting is an extremely challenging one for both trajectory-based and density-based importance sampling since the behavior and evaluation policies are so far apart. However, even in this extremely difficult setting, the there exists an interpolating estimator within the W-SOPE$_n$ spectrum that is able to have better performance than either of the endpoints.