# OpenReview forum: "SOPE: Spectrum of Off-Policy Estimators"
_NeurIPS.cc/2021/Conference — NeurIPS 2021 Poster_

### Official Review · Reviewer_EWnS · 2021-07-06

**Rating:** 5
**Confidence:** 4

**Summary:**

The scope of the paper is within off-policy evaluation (OPE) in reinforcement learning (RL). The paper aims to develop a spectrum of OPE methods which interpolate between trajectory based importance sampling (IS) and stationary state distribution IS.

**Limitations And Societal Impact:**

At the moment, OPE papers of this sort are mostly theoretical and therefore have limited societal impact. The long term goal of OPE work is to allow deployment of RL policies in risk sensitive domains, which would have negative impact if not done properly. The authors acknowledge the importance of OPE in that regard and thus satisfy the criterion for discussing the potential negative impact of their work.

**Main Review:**

To briefly summarize my review and justify my score, I believe the paper is wrong. The method introduced relies on Lemma 1 which as far as I can tell is not correct. This perhaps leads to some inconsistent results which the authors do not address in the experimental section. Having said that, I think the idea of generating a family of estimators interpolating between IS and SIS is a good one, and I will therefore do my best to provide feedback which will be useful in the future if the authors can find a mathematically consistent way to generate such estimators.

The estimators presented in the paper rely on the derivation of equation (1), which uses Lemma 1. I believe the Lemma is wrong. An easy way to see that the Lemma is probably wrong is that the LHS is independent of \gamma and L and only depends on the MDP dynamics and policies, while the density ratios on the RHS do depend on \gamma. As a quick counter example to demonstrate it, think of a deterministic two state system, which always starts at s0, with two actions (a0 and a1). The agent always starts at s0, from which a0 leaves the agent in s0 and a1 transitions to the absorbing state s1. Assume a state independent evaluation/behavior policy which has probability \pi_{e/b}(a0) to take action a0. To compute the LHS of lemma 1 for S_z = s0 (ignoring actions for now), we note that if S_z = s0, it means a0 was always taken, and therefore E[\rho_{1:z} | S_z = s0] = (\pi_{e}(a0) / \pi_{b}(a0))^z. On the other hand the RHS can be shown to be equal \frac{\sum_t^L \gamma^t \pi_{e}(a0)}{\sum_t^L \gamma^t \pi_{e}(a1)}. Note that these expression may differ slightly depending on how to interpret an inconsistency the authors have in the notation and definition which I will detail later. These quantities are obviously different. Looking at the actual proof in the appendix, one place where it is wrong is in their use of the term P_{e/b}(S_z, A_z) which is not well defined as the probability of the MDP being in (S_z, A_z) is a function of time which is absent in the derivation. In their proof and justification of a later part of the derivation the authors invoke explicitly incorporating the time into the state which might help here. If you do that you can maybe make the claim that \gamma = 1, and get rid of that dependency, but as the counter example shows you will still not get the Lemma 1 equivalence. More importantly, the inclusion of that assumption (time explicit in the state) is misleading. While it is possible to do it, it is a very uncommon practice in RL (usually for the episodic case you just have the episode end at a given time but have time dependent value function/policies), and if the authors make that very unusual and restrictive assumption it should not be hidden at the end of the appendix.

Based on this error alone, I do not believe the paper is publishable at this point. Additionally, while I do not expect the empirical results to be valid given the error, I think there are several inconsistencies in the results which can suggest that something is wrong:
Fig 3 and 4 top middle : if n=0 corresponds to SIS and n=L to PDIS the bias should be monotonically decreasing with n. Instead it is not monotonic in Fig 3 and sometimes monotonically increasing in fig 4.
Fig 3 and 4 bottom right : Why for some parameter (# of trajectories or epsilon) the variance increases with n and for some decreases, where the logic of interpolation between SIS to PDIS would predict monotonic increase for all.

Some other comments:
- The definition of J(\pi) and d^\pi is inconsistent between the first paragraph of the "notation" section in the background and the "density based IS" section (indexing of t).
- Maybe I missed it, but CWPDIS is never discussed before it is given before eq. 3, and is not cited. While it would be familiar to people who know the OPE literature well, it is not used commonly enough to appear without citation. On that note, it is a little ironic that eq. 3 appears as an analogue of CWPDIS without proof of consistency, since CWPDIS was originally introduced by Pillip Thomas of an example of how naively meshing two estimators (in the case presented in his thesis meshing PDIS and WIS into what was then known as WPDIS) can easily result in inconsistent estimators.
- In any future work, I would strongly encourage the authors to include a continuous environment in their experiments. One of the main failing mechanisms of SIS compared to IS is that any type of non-markovianity (or partial observability) would lead to inconsistent estimators. Meanwhile such partial observability doesn't effect IS as long as the policies are Markovian. Thus, it would be interesting to see how interpolation between SIS and IS can work in continuous domains where a form of partial observability is introduced through function approximation.

Last, I would like to point out that the paper has too many typos for what I feel is acceptable in a submission.

**Time Spent Reviewing:**

6 to 8

---

> ### Author Response · Authors · 2021-08-10
> **Response to Reviewer 4**
>
> Thank you for your feedback! We appreciate the attention to detail and careful checking of the correctness of our work. Your comments helped us realize we could have presented the derivation of our estimator more clearly to avoid any confusion to readers.
>
> **Infinite and Finite Horizons**
>
> While most density based OPE methods are presented solely in the infinite horizon setting, we decided to keep the presentation of our estimator general and not exclude the finite horizon setting. As rightly mentioned by the reviewer, time dependent value function/policies are often used in the episodic setting. However, note that adding time to the state representation does not create any additional restriction when the functions that use that state are already dependent on time. Thus, to maintain the Markov assumption for all finite horizon MDPs and to unite the notation for the finite and infinite horizon version of our estimator, we included time in the state for the finite horizon setting. We do not include time in the state for the infinite horizon setting.
>
> Specifically, since time is included in state in the finite horizon setting, we used $\frac{d^{\pi_e}}{d^{\pi_b}}$ for both finite and infinite horizon cases to give a single estimator for both settings with unified notation. We see that this choice may lead to some confusion, and that we can be more clear by explicitly writing $\frac{d^{\pi_e}}{d^{\pi_b}}$ in the infinite horizon version and $\frac{d_t^{\pi_e}}{d_t^{\pi_b}}$ in the finite horizon version of the estimator instead of adding time to state in the finite horizon. We will also emphasize above Eqn 2 that the time is not included in the state for the infinite horizon setting.
>
> **Lemma 1 and Counterexample**
>
> > “To briefly summarize my review and justify my score, I believe the paper is wrong. The method introduced relies on Lemma 1 which as far as I can tell is not correct.”
>
> We will go into the specifics about the correctness of Lemma 1 below, but first we would like to  emphasize that our method is not solely reliant on Lemma 1. Particularly, for the infinite horizon case, Appendix B provides an additional derivation that does not use Lemma 1. Rather, this derivation relies on using the Bellman recursion of the average visitation distributions $d^\pi(s,a) = (\sum_{t=0}^\infty \gamma^t d_t^\pi(s,a)) / (\sum_{t=0}^\infty \gamma^t)$. In the infinite horizon setting, we do not need to include time in the state to maintain the Markov property, and so in this case the $d^\pi(s,a)$ in the estimator gives the discounted average visitation to state $s$ and action $a$ across all time. Note that the infinite horizon case allows us to interpolate between IS likelihood ratios and visitation ratios $d^{\pi_e}(s,a) / d^{\pi_b}(s,a)$ that can be estimated by methods like the DICE family.
>
> > “An easy way to see that the Lemma is probably wrong is that the LHS is independent of \gamma and L and only depends on the MDP dynamics and policies, while the density ratios on the RHS do depend on \gamma”
>
> We will emphasize around Line 117 that Lemma 1 is presented in the context of the derivation for the finite horizon MDP with time included in the state. For this setting, $d^\pi((S_z, z), A_z)) = \gamma^z d_z^\pi((S_z, z), A_z)$ because for all $t \neq z, d_t^\pi((S_z, z), A_z) = 0$. We can see then that when we consider the density ratios, the $\gamma^z$’s will cancel out, so both LHS and RHS do not depend on $\gamma$.
>
> To see this more specifically, we apply this to your counterexample. (To avoid overloading subscripts for timestep, we write a0 and a1 for your domain as x and y.) As you have stated, LHS is equal to $\mathbb{E}[\rho_z | S_z, A_z] = \left(\frac{\pi_e(x)}{\pi_b(x)}\right)^z$. However, since in the finite horizon setting we include time as a part of state, the RHS is equal to $\frac{d^{\pi_e}((S_z, z), A_z)}{d^{\pi_b}((S_z, z), A_z)} = \frac{\sum_{t=1}^L \gamma^t d_t^{\pi_e}((S_z, z), A_z)}{\sum_{t=1}^L \gamma^t d_t^{\pi_b}((S_z, z), A_z)} = \frac{\gamma^z d_z^{\pi_e}((S_z, z), A_z)}{\gamma^z d_z^{\pi_b}((S_z, z), A_z)} = \frac{d_z^{\pi_e}((S_z, z), A_z)}{d_z^{\pi_b}((S_z, z), A_z)} = \left(\frac{\pi_e(x)}{\pi_b(x)}\right)^z$. Thus, neither side depends on $\gamma$ and LHS=RHS, as needed.
>
> **Consistency and CWPDIS**
>
> > “On that note, it is a little ironic that eq. 3 appears as an analogue of CWPDIS without proof of consistency, since CWPDIS was originally introduced by Pillip Thomas of an example of how naively meshing two estimators (in the case presented in his thesis meshing PDIS and WIS into what was then known as WPDIS) can easily result in inconsistent estimators.”
>
> Thank you for catching that we left out formally introducing CWPDIS in our background section. We will make sure to update the background to include definition.
>
> We chose not to include consistency claims because the focus of this paper is on introducing the interpolation and using it to understand the relationship between IS and SIS. In future work, we plan to explore more how to utilize this interpolation practically. For instance, we would like to further investigate how to best choose $n$ or combine the estimates for different $n$. We believe that consistency claims would be better presented alongside those results. For example, if we consider a practical algorithm that selects $n$ for WNSTEP such that $n$ increases with more data, it is trivial to see that when $n$ converges to $L$ the WNSTEP estimator converges to CWPDIS and thus will be a consistent estimator even if the SIS estimate is not consistent.
>
> **Experimental Results**
>
> > “I think there are several inconsistencies in the results which can suggest that something is wrong”
>
> Thank you for the careful checking of our experimental results. We believe that the concerns you have can be addressed by noting that our experiments use WNSTEP which interpolates between CWPDIS and weighted-SIS. As discussed in the paragraph at line 161, while WNSTEP does practically perform a bias-variance tradeoff, the tradeoff is not perfectly clean due to the bias in CWPDIS.
>
> The main intention of our experiments was to showcase that there exists scenarios where an estimator within the spectrum can outperform the IS and SIS endpoints, and we demonstrate this. From your feedback, we see that we can make an even stronger point about the precise bias-variance tradeoff of our method by also including the results from the non-weighted version. We will include such experiments in the final version of our paper.
>
> > “Fig 3 and 4 top middle : if n=0 corresponds to SIS and n=L to PDIS the bias should be monotonically decreasing with n. Instead it is not monotonic in Fig 3 and sometimes monotonically increasing in fig 4.”
>
> Since we interpolate between CWPDIS and weighted-SIS for our experiments, we do not expect to see bias to be perfectly monotonically decreasing with n. We further note that the top graphs *do* demonstrate that our method has a bias-variance tradeoff, even if the tradeoff is not as exact as if we used an unweighted version of the estimator.
>
> In the Fig 3a bias plot, we show the experimental results from varying the size of the historical dataset. From the plot, we can see that the $n=0$ case corresponding to SIS has the most bias and interpolating estimators have less bias. This shows there exist scenarios where an interpolating estimator can be less biased than SIS.
>
> In the Fig 4a bias plot, we show the experimental results from varying the closeness of the behavior policy and evaluation policies. In particular, we fix $\pi_e=0.9$ and vary $\pi_b$ from 0.1, 0.3, 0.5, 0.7. Since CWPDIS performance is heavily impacted by the amount of policy mismatch, it is not surprising to see that CWPDIS can also be extremely biased in the cases where the divergence is high. In such cases, it makes sense that there may not exist an estimator within the spectrum that can outperform the endpoints and the results support that. However, when the policies are not too far like when $\pi_b = 0.7$, the bias of the CWPDIS estimator is less than that of the SIS estimator, and we are able to tradeoff bias and variance to get an estimator within the spectrum that outperforms the endpoints.
>
> > “Fig 3 and 4 bottom right : Why for some parameter (# of trajectories or epsilon) the variance increases with n and for some decreases, where the logic of interpolation between SIS to PDIS would predict monotonic increase for all”
>
> Since we are interpolating between CWPDIS and weighted-SIS, we do not expect to see a perfectly monotonic increase in variance. In particular, we recall that CWPDIS normalizes the likelihood ratio weights to reduce variance of the IS estimate. Additionally, following a similar analysis like above, we can see the bottom graphs *do* demonstrate a bias-variance tradeoff, even if it is not perfectly monotonic.
>
> **Continuous Experiments**
>
> > “In any future work, I would strongly encourage the authors to include a continuous environment in their experiments.”
>
> Thank you for the suggestion! We agree that we would have been able to even further showcase the benefits of our method if we had included a continuous environment in our experiments. In particular, with more bias in our SIS estimate, the bias-variance tradeoff of the interpolation would be even more pronounced.
>
> We were able to run larger experiments on the standard continuous mountain car benchmark after the deadline, and we were able to confirm that the interpolation had this predicted bias-variance tradeoff in this setting. We will include the results in the final version of the paper.
>
> Thank you for other suggestions as well! We will incorporate those and fix all typos. Finally, we would also like to thank you for the supportive comments regarding the overall idea of interpolating estimators.

---

> > ### Comment · Reviewer_EWnS · 2021-08-25
> > **Response to authors' feedback**
> >
> > I would like to thank the authors for a thorough and detailed response.
> >
> > The authors response to my concern regarding the correctness of Lemma 1 was convincing, but I think the authors must highlight many times throughout the paper the explicit inclusion of time in the state, as it is not a common formulation and the correctness of the paper relies on it.
> >
> > Regarding the authors' explanations to the experiments, I am partially satisfied. I agree with the authors' comment that the fact that the experiments interpolate between weighted versions of SIS and PDIS makes decoupling bias and variance tradeoffs difficult. However, I think the authors should do a better job of trying to entangle it anyway. One way in which they could do that is preform similar experiment with the non-weighted versions of the estimators so there is non confounding bias effects from weighting (I can imagine the non-weighted versions would have huge variance which would make the signal too noisy, but that can be overcome by using larger number of trajectories). Also, it would be great if the authors can explain the huge variance reduction for the graph domain between n=0 and n=1.
> >
> > I personally think the paper could use another round of improvement before publication, but after reading the authors' feedback my objections to its publication are not strong and I will change my score to only a marginal reject, especially given that its overall idea is a very nice one.

---

### Official Review · Reviewer_eekS · 2021-07-14

**Rating:** 8
**Confidence:** 4

**Summary:**

This paper introduces a novel n-step method which interpolates between PDIS ($n=H$) and density estimation ($n=0$) methods (the DICE family). The paper derives the relationship between density ratio corrections and likelihood ratio corrections, leading to the development of the proposed algorithm. The paper empirically investigates multiple ablation studies of the bias/variance tradeoff of the proposed algroithm.

**Limitations And Societal Impact:**

Limitations and societal impact are sufficiently addressed.

**Main Review:**

I recommend to accept this paper. [S1] The connection between density ratio and likelihood ratio corrections is insightful. [S2] The proposed algorithm is simple and provides an avenue to avoid a common criticism of DICE methods: the difficulty in estimating the density ratio and resultant bias. [S3] The empirical evaluation is motivating. Finally, [S4] the paper is quite well-written and very clear.

I have only very minor reservations: [W1] the empirical investigation exclusively uses weighted-PDIS and weighted-SIS, so interpolates between two biased methods. [W2] By using small toy domains where SIS can perfectly estimate the density ratio, many  of the stated difficulties of SIS are not investigated, leaving open-ended the contribution of this paper in harder problem settings.

---

**[W1]**: Weighted estimators

I would be curious to see a more clear demonstration of the bias-variance tradeoff by additionally investigating the non-weighted versions of these estimators---at the very least a non-weighted version of PDIS. The bias on the graph MDP is sufficiently small for all values of the estimator, that I wonder if the empirical estimate of bias is more noise than signal (Figure 3a), but I would have expected to see the bias decay more strongly as n -> L. In the case of Figure 4a, I am surprised that the PDIS estimate _increases_ bias over the weighted-SIS estimate leaving the bias-variance tradeoff claims poorly supported.

---

**[W2]**: Small domains

I generally don't mind small domains. I believe they were harmful in this case where the true density ratio is representable within the function approximation class. Using even the toy Mountain Car domain with a reduced function approximator would provide some additional value by demonstrating the impact of the proposed n-step estimator when SIS has much worse bias. Currently the best value of $n$ is often very close to 0 (near SIS), but in larger/more realistic settings I wonder if this would continue to be the case.

---
Minor question: would it be possible to derive an expression for minimizing the MSE w.r.t. parameter $n$ analytically? I wonder if this could be estimated easily from the data. At the very least, I would be curious to see how effectively an n-fold cross-validation approach selects the optimal value of $n$.

---

**Edit after discussion period:**

After concerns about the correctness of the proof have been resolved, I continue to feel this paper is a solid contribution to the field. My previous concerns were only minor in strength and have been well-addressed with the author response and I do not feel that I would need to re-review after the new experiments are added to the paper. I recognize the discussion period was rather quiet for this paper, but frankly, I believe this paper left little need for added discussion---to me it is a clear accept.

**Time Spent Reviewing:**

6

---

> ### Author Response · Authors · 2021-08-10
> **Response to Reviewer 3**
>
> Thank you for your support and feedback!
>
> > W1: “I would be curious to see a more clear demonstration of the bias-variance tradeoff by additionally investigating the non-weighted versions of these estimators”
>
> Thank you for the suggestion! Our original goal for our experiments was to simply demonstrate that there existed scenarios where an interpolation within the spectrum could outperform the endpoints IS and SIS.
>
> We see from your suggestions that our experiments can be more impactful if we also use a non-weighted version of the estimator, allowing us to give an even more precise investigation of the exact bias-variance tradeoff. We also include results from a non-weighted version of the experiments in the final version of our paper.
>
> > W2: “By using small toy domains where SIS can perfectly estimate the density ratio, many of the stated difficulties of SIS are not investigated, leaving open-ended the contribution of this paper in harder problem settings.”
>
> > “Currently the best value of n is often very close to 0 (near SIS), but in larger/more realistic settings I wonder if this would continue to be the case.”
>
> Thank you for the feedback! We completely agree that including more complex domains where density estimation exhibits more bias would help us to even better demonstrate the impact of our method.
>
> We were able to finish running some larger experiments on the continuous Mountain Car benchmark after the deadline. Just as you predicted, in this setting where we had a much more biased estimate of the density ratio, our results showed that the optimal $n$ tended to be larger. This further demonstrates clearer evidence of the bias and variance tradeoff of our method. We will include the full results from these experiments in the final version of this paper.
>
> > “would it be possible to derive an expression for minimizing the MSE w.r.t. parameter n analytically? I wonder if this could be estimated easily from the data. At the very least, I would be curious to see how effectively an n-fold cross-validation approach selects the optimal value of n.”
>
> Thank you for the suggestion! We haven’t yet looked into an expression for minimizing MSE, but it sounds like promising future work. Our original intention for this paper was to just introduce the interpolation and use it as a way to understand the relationship between IS and SIS. Thus, we originally intended to leave more practical issues about using the interpolation like $n$ selection and blending the different $n$ estimates for future work.
>
> However, we see from your suggestion that it would be helpful to include some basic guidance for using the practical $n$ selection. We’ll take a look into an MSE expression, and we’ll include some basic experiments on how well cross-validation can be used to select $n$.

---

### Official Review · Reviewer_TiwR · 2021-07-20

**Rating:** 7
**Confidence:** 4

**Summary:**

The paper proposes an estimator for off-policy evaluation (OPE) that unifies trajectory-based per-decision importance sampling with stationary distribution importance sampling using an n-step approach analogous to multistep temporal-difference learning. The paper presents n-step estimators for both ordinary importance sampling and weighted importance sampling, and investigates their properties on two tabular OPE environments.

**Limitations And Societal Impact:**

I was satisfied with the paper's discussion of limitations and societal impact.

**Main Review:**

I recommend publishing the paper at NeurIPS 2021 despite minor concerns about scope.

**Quality:**

The proposed estimator is derived in a principled fashion, and the experiments are aimed at understanding properties of the proposed estimator and seem well-done to me.
The main thing that disappointed me about the paper was that it restricted itself to n-step methods instead of also doing the obvious next step and incorporating exponential weighting to get an estimator analogous to TD($\lambda$).
A secondary concern is that there is no guidance for how to set the "switching time" parameter for an environment a priori.
In general I'm not a fan of introducing parameters, but the paper appears to do it for the purposes of better understanding two existing algorithms instead of doing it to improve performance, so I'm ok with it.
One could also argue that another experiment in a larger domain is needed, but I was satisfied with the existing experiments.

**Clarity:**

The paper is written clearly and is very easy to understand. There are some minor grammatical errors throughout (some of which I've pointed out below). The paper doesn't try to obfuscate or dress up what it's doing, which is frankly a breath of fresh air.

**Significance:**

The proposed method provides a useful unification of two existing algorithms (useful because intermediate values can often perform better than either extreme), which could easily be built upon. Interestingly, the n-step approach forms a bridge to multi-step TD learning, which provides another perspective on the problem.

**Originality:**

The paper takes a well-known technique (n-step estimates) and applies it to a different problem. To the best of my knowledge the proposed method is novel.

**Misc. comments, questions, and suggestions for improvement:**

Line 29: There's an extra "can" in this sentence.

Line 31: "If" should be lowercase.

Line 42: "the first part" is repeated.

Line 43: "using" should be uppercase.

Line 69: It was a little confusing to define $c=\sum_{t=0}^{L} \gamma^t$ instead of just writing it out.

Line 146: "to" is repeated.

Line 151: should be "allow us to control the variance's dependency"

Line 162: "amount" is repeated.

Figure 3: Thank you for plotting confidence intervals! I can easily see whether differences are statistically significant by checking if the shaded regions overlap.

Line 160: The acronym CWPDIS is never actually defined anywhere that I could find. I'm guessing it's Conditional? Weighted Per-Decision Importance Sampling?

**Time Spent Reviewing:**

7

---

> ### Author Response · Authors · 2021-08-10
> **Response to Reviewer 2**
>
> Thank you for the support and the suggestions!
>
> > “The main thing that disappointed me about the paper was that it restricted itself to n-step methods instead of also doing the obvious next step and incorporating exponential weighting to get an estimator analogous to TD(λ)”
>
> > “A secondary concern is that there is no guidance for how to set the "switching time" parameter for an environment a priori. In general I'm not a fan of introducing parameters, but the paper appears to do it for the purposes of better understanding two existing algorithms instead of doing it to improve performance, so I'm ok with it.”
>
> Thank you for the feedback! Our original intention for this paper was to focus on introducing the foundational idea that IS and SIS lay as endpoints on a spectrum of estimators, and explore how the spectrum naturally allows us to understand the bias-variance tradeoff relationship between IS and SIS.
>
> While this paper is focused on exploring the most foundational aspects of this spectrum, we are excited to build upon this foundation in future works to further understand and improve off-policy evaluation, including by pursuing the natural follow-ups mentioned like $n$ selection and TD($\lambda$).
>
> > One could also argue that another experiment in a larger domain is needed, but I was satisfied with the existing experiments.
>
> Thank you for your support! Regarding additional experiments, we were able to finish running experiments on the standard continous Mountain Car benchmark after the deadline. The results of these experiments further illustrated that our interpolation method was also able to tradeoff bias and variance and outperform IS and SIS on continuous domains. The full results from these experiments will be included in the final version of our paper.
>
> Thank you also for your additional suggestions for improvement! We will address them all in the final version of our paper.

---

### Official Review · Reviewer_VgQT · 2021-07-21

**Rating:** 7
**Confidence:** 5

**Summary:**

The paper considers the problem of off-policy evaluation and shows that it is possible to interpolate between the two common families of algorithms: importance sampling of policies which is unbiased but high variance, and correcting state distributions, which is low variance but could be very biased since the distributions need to be estimated. The resulting algorithm is tested empirically on small tabular tasks and it is shown that there are indeed intermediate values in the spectrum that achieve better MSE than either of the extremes.

**Limitations And Societal Impact:**

Yes

**Main Review:**

This paper was a pleasure to read. Despite not having strong experiments, the key idea is very cool and advances off-policy research in a clear way, and the paper is written very well. I only have a couple of high-level comments / questions, and some minor proofreading remarks.

1) Figure 1 intuition is important but gets lost afterwards. It would be great to show that the error is lowest when split at the hallway, but for that z would need to depend on the state. Can the authors provide an intuition about this?
2) Larger experiments are promised (optimistically!) in the appendix but not found. I think smaller experiments are fine, but it would be nice to have more insight about the trends. For example it seems that when the policies are further apart n=1 is optimal (single importance sampling ratio). Do the authors have intuition of what this would look like in practice?

Minor comments
* Lines 29-30: something funky with the phrasing
* 68: d^pi_t is usually phrased as probability which might be clearer. Also I guess it should be s_t, a_t
* Big equation on page 2: Something off with the indices here — shouldn't the sum start at t=0 (since numbering rewards starts from 0 in tau)?
* Also, shouldn’t d_0 be a part of the definition of d^pi for probability of (s,a) at the initial time step?
* 115: third I think? Marking the relevant equality would be clearer
* Figure 3(b): why does variance decrease for the intermediate amounts of trajectories?
* 159: Maybe named paragraph here?
* 160: CWPDIS mentioned for the first time
* 162: amount amount
* Figure 4: it would be clearer to denote lines by policy distance instead, since the targets are different for the two tasks
* 181-182: No larger continuous state spaces in the appendix :(
* 198: what is H here?
* 200: has has

**Time Spent Reviewing:**

3

---

> ### Author Response · Authors · 2021-08-10
> **Response to Reviewer 1**
>
> Thank you for the encouragement and feedback!
>
> > 1. “Figure 1 intuition is important but gets lost afterwards. It would be great to show that the error is lowest when split at the hallway, but for that z would need to depend on the state. Can the authors provide an intuition about this?”
>
> Thank you for the suggestion! We agree that the intuition from Figure 1 is very helpful to understanding our estimator, and we will incorporate more discussion alongside our derivations to further incorporate this intuition. As for allowing z to depend on state, one way to do this is to allow z to be a random variable denoting the time step the agent enters the hallway state.
>
> > 2. “Larger experiments are promised (optimistically!) in the appendix but not found. I think smaller experiments are fine, but it would be nice to have more insight about the trends. For example it seems that when the policies are further apart n=1 is optimal (single importance sampling ratio). Do the authors have intuition of what this would look like in practice?”
>
> Thank you for the feedback! Regarding the promised non-tabular experiments, we were able to finish running experiments on the standard continuous Mountain Car benchmark after the deadline. The results showed that our interpolation method was also able to trade-off bias and variance and outperform IS and SIS methods on continuous domains. We will include these full results in the final version of our paper.
>
> We will also include more in depth analysis on the trends for all our experimental results. As for why when policies are further apart n=1 is optimal in our experiments, we believe this is because the variance from the product of action likelihood ratios can dominate the bias from SIS when policies are very different, so keeping a small $n$ turns out to be better for MSE.
>
> Thank you also for the comments about fixes to improve the readability of our paper! We will address all of these in the final version.

---

### Public Comment · Authors · 2021-12-29
**Camera Ready**

Thank you reviewers again for your feedback! We took your suggestions into consideration in the camera ready version of our paper. The changes we made were:
1. Inclusion of a marginalization step over time in the presentation of the derivation of the SOPE estimator (see Appendix B). This additional step explicitly unifies the finite and infinite horizon version of our estimator, and resolves the issues raised by Reviewer 4 regarding embedding time into state
2. Inclusion of additional experimental results of the non-weighted version of the  estimator, based on suggestions from Reviewers 3 and 4 (see Figure 4)
3. Addition of doubly robust extension of the SOPE estimator that was not included in the initial paper (see section 5)
4. Update algorithm’s name from NSTEP/WNSTEP to SOPE/W-SOPE

---

### Decision · Program_Chairs · 2021-09-27

**Decision:**

Accept (Poster)

**Comment:**

The paper considers the problem of off-policy evaluation and shows that it is possible to interpolate between the importance sampling estimator, which is unbiased but high variance, and correcting state distributions, which is low variance but could be very biased since the distributions need to be estimated. The authors showed that the optimal estimator with lowest MSE is indeed taking an intermediate values in the spectrum to mix these two estimators.

The proposed estimator is derived in a principled fashion, the results are insightful, and the paper is very well written. Generally this paper's contribution are non-trivial yet interesting, and it can definitely be a valuable addition in the vast literature of off-policy evaluation in RL. Some potential drawback of this paper includes simplistic experiments that are aimed only for proof-of-concept, and the restriction to to n-step estimators.  During the discussion phase, the reviewer also recommends doing another round of improvement before publications to study the confounding effect of these two estimators. Please try to do so to improve the (already good) quality of this work.

In general I believe the merits of this work surpass the improvements required. So I also recommend acceptance.